# Advancing interpretation of incoherent scattering in ice penetrating radar data used for ice core site selection

Ellen Mutter[1] and Nicholas Holschuh[2]

[1]Department of Earth and Atmospheric Sciences, Cornell University, Ithaca, NY, 14853, USA
[2]Department of Geology, Amherst College, Amherst, MA 01002, USA

*Correspondence to*: Nicholas Holschuh (nholschuh@amherst.edu)

**Abstract.** Below the coherent layering in ice penetrating radar data collected in Antarctica and Greenland, incoherent scattering is common. This scattering is signal, not noise, and has the potential to inform our understanding of the structure and dynamics of the bottom 20% of glaciers and ice sheets. Here, we present a comparison between radar imagery and ice core properties for sixteen ice core sites across Antarctica and Greenland, to identify possible sources for incoherent scattering and evaluate its use in ice core site selection. We find that incoherent scattering is commonly coincident with either gradual changes in crystal orientation fabric or rapidly fluctuating fabrics in deep ice, where strain is localized by strength differences associated with ice grain size. Macro-scale deformation and layer folding at scales below the range-resolution of radar does not seem to result in incoherent scattering or induce an echo free zone, as has been previously hypothesized. Where incoherent scattering is laterally homogeneous in intensity, layering is typically undisturbed in nearby ice cores. But where incoherent scattering is laterally heterogeneous in intensity and the pattern does not appear conformal with subglacial topography, we find multi-meter-scale folding and associated discontinuities in nearby ice core records. Future higher-resolution sampling of fabric in ice cores would allow for more quantitative interpretation of incoherent scattering and its amplitude, but we show that the qualitative nature of incoherent scattering has the potential to inform us about the continuity of climate records at prospective ice core sites and should be considered when evaluating the nature and quality of basal ice.

## 1. Introduction

Existing ice cores provide our best record of past atmospheric chemistry. These cores capture global climate changes over the Holocene and Late Pleistocene (Wolff et al., 2010). Future ice coring initiatives hope to build on that record, both extending it further back in time (Jouzel and Masson-Delmotte, 2010) and measuring regional climate change (Mulvaney et al., 2021) during specific climate periods (Fudge et al., 2023). These future projects focus on the identification and collection of very specific ice, and so they typically start with extensive geophysical surveying for "site selection" preceding drilling. Ice penetrating radar data have served as the primary tool for this work, which uses layering in radar imagery to infer spatially variable accumulation, basal melting, and ice flow, and through that, identifying ideal ice core sites (Bingham et al., 2024; Chung et al., 2023, Karlsson et al., 2018; Schroeder et al., 2020). But site selection has relied primarily on the strong, coherent signal that spans the upper three-quarters of the ice column in most radar imagery. Here we focus on improving interpretation

of other signals in radar data, with a particular focus on what deep incoherent scattering (described in section 2) can tell us
about ice near the ice sheet base.
All radio-wave scattering in ice originates from dielectric contrasts. To better understand the nature and sources of scattering
in existing ice penetrating radar data, several previous studies have compared radar imagery to observations of ice chemistry
and physical properties measured in ice cores (e.g., Eisen et al., 2003, 2007; Hammer, 1980; Harrison, 1973; Millar, 1982;
Mojtabavi et al., 2022). But that work has focused on the coherent, isochronal layering, and comparatively little has been done
to understand the deeper signals, which are becoming better sampled with modern, high power / low noise systems. This deep
ice has also become increasingly scientifically important, as it is at the center of the search for an ice core record that spans
the Mid-Pleistocene transition (Chung et al., 2023; Lilien et al., 2021). Using data from 16 deep ice cores across Antarctica
and Greenland (Fig. 1), we work to better understand the physical properties that produce deep, incoherent scattering, and
evaluate the extent to which it may be diagnostic of layer disturbances or other disqualifying characteristics when pursuing
future ice cores.

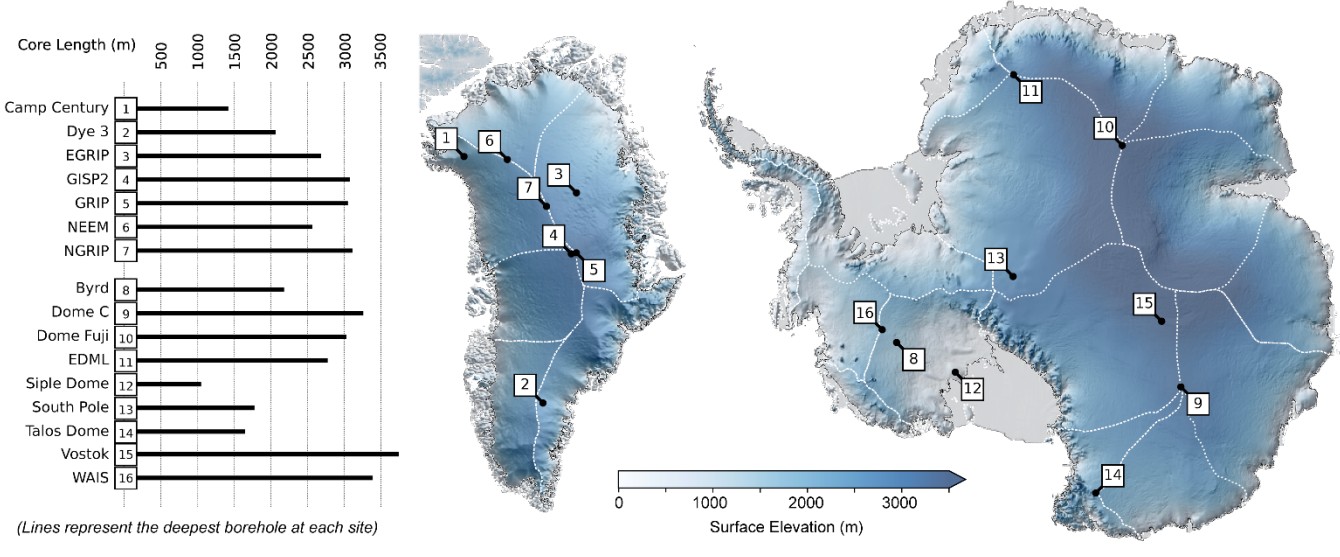


**Figure 1: Locations of deep ice coring initiatives in Greenland and Antarctica used in this study and the lengths of the associated**
**cores. Surface elevation maps of Antarctica** (Howat et al., 2019) **and Greenland** (Porter et al., 2018) **with catchment boundaries**
(Mouginot and Rignot, 2019; Rignot et al., 2013) **showing ice divides in white.**

## 2. Background: Scattering and the Radar Imaging Problem

Radar systems actively transmit energy into the subsurface. Time-of-flight measurements for back-scattered energy (together with a known speed of light in ice) can be used to infer the position of subsurface scatterers and reconstruct the geometry of glacier systems (Bingham et al., 2024; Dowdeswell and Evans, 2004). In the near sub-surface, contrasts in the dielectric permittivity that scatter energy are controlled primarily by variations in density, while most deeper englacial reflectors arise from either conductivity contrasts, due to variations in the concentration of free ions deposited with the snow at the surface (Stillman et al., 2013), or transitions in the ice crystal fabric, typically localized by changes in grain size also arising from impurity deposition (Fujita et al., 1999). Fabric induced scattering is a product of the dielectric anisotropy of individual ice crystals, with transitions in c-axis fabric capable of producing an (up to) ~1.3% contrast in the polarization-dependent bulk permittivity (Matsuoka et al., 1997). Incoherent scattering may come from both chemical and physical sources; we work to provide some of the first constraints on its origins here.

Glaciologists primarily use radar data for ice core site selection in two ways. The first approach is focused on the geometry of coherent, isochronous layering within the ice sheet (an example of which can be seen in the upper portions of Fig 2.a). These layers originate as flat-lying layers of snow at the ice sheet surface and are transformed by flow during burial; thus, their geometry can be used to diagnose spatial variations in accumulation (e.g., Karlsson et al., 2020), glacier sliding (e.g., Leysinger Vieli et al., 2007), and basal melt (e.g., Bingham et al., 2024; Fahnestock et al., 2001). The second approach is focused on the nature of subsurface scattering, both its coherence (e.g., Lindzey et al., 2020; Oswald et al., 2018; Schroeder et al., 2015) and amplitude (e.g., Catania et al., 2003; Christianson et al., 2016; Chu et al., 2018), which together can be used to infer the modern electrical (and, more generally, material) characteristics of the ice sheet and its substrate.

Subsurface targets can be divided into two main categories: specular interfaces and rough (or diffuse) scatterers (Schroeder et al., 2015). Specular interfaces, like mirrors, scatter energy in one dominant direction, a function of the direction-of-arrival for the incoming radio wave and the orientation of the interface. Diffuse scatterers redistribute incident energy at a variety of angles. This leads to significant differences in the coherence of the scattering between specular and diffuse targets (defined here as the consistency in phase and amplitude of the backscattered energy with slight changes in the position of the radar system). Incoherent scattering typically occurs at rough interfaces or when there are multiple diffuse scattering targets at a similar range from the instrument. It has been observed as a product of rare glacier conditions, for example, where there is significant temperate ice and associated englacial water (Hamran et al., 1996) or where debris has been entrained near the base of glaciers (Winter et al., 2019). But it must also be generated by more common glaciological phenomena, as it is present within several hundred meters of the ice sheet base across large parts of Antarctica and Greenland.

Consider the example radar image in Fig. 2.a. Each pixel represents either backscattered energy or electrical or thermal noise
in the radar electronics. The position of the radar system varies across the columns in the image, and the delay-time following
the transmitted pulse (associated with the range to possible targets) varies across the rows in the image. In regions dominated
by planar, specular interfaces (as in the upper half of Fig. 2.a), each pixel typically represents backscattered energy from only
a single direction of arrival. This is because, even though there are many scattering targets at the range associated with that
pixel (as shown in Fig. 2.b), only that interface tangential to the range shell (such that the interface is normal to the propagating
wave) returns energy to the system. But in regions where there are diffuse scatterers, each pixel in a radar image represents the
interference of scattering from multiple targets, with backscattering arriving from multiple angles (Fig. 2.b.ii/iii). With slight
changes in the position of the system, the dominant source of scattering at a given range can change, resulting in little
consistency in phase or amplitude from pixel to pixel. This is extremely common for energy arriving below the ice bottom
reflector, with a long tail of incoherent scattering appearing at greater range (Fig. 2.a.iii). Less well described is incoherent
scattering from within the ice column (Fig. 2.a.ii) which is the focus of our research here.

When considering the nature of scattering in radar imagery, it is important to remember that the images themselves are
ultimately a product of three things:

   1.   The geometry and physical / electrical characteristics of the glacier subsurface.

   2.   The system used to collect the data (including the characteristics of the transmitted wave, antennas, and transmit /
        receive electronics).

   3.   The filtering, focusing, and additional image processing algorithms applied after collection.

The nature of radar targets depends on both the scale of electromagnetic heterogeneity in the medium and the frequency content
of the transmit pulse (with higher frequencies / bandwidths associated with finer range resolution). This is because the
specularity of a target is ultimately dictated by the Rayleigh roughness criterion for an interface, with specular scattering
occurring when roughness elements are less than $1/8^{th}$ the scale of the dominant radar wavelength (Peters et al., 2005). Figure
2.c demonstrates how the same targets manifest differently across different radar systems; with lower resolution systems,
scattering appears more structured, like the specular and coherent layering in the shallow ice.

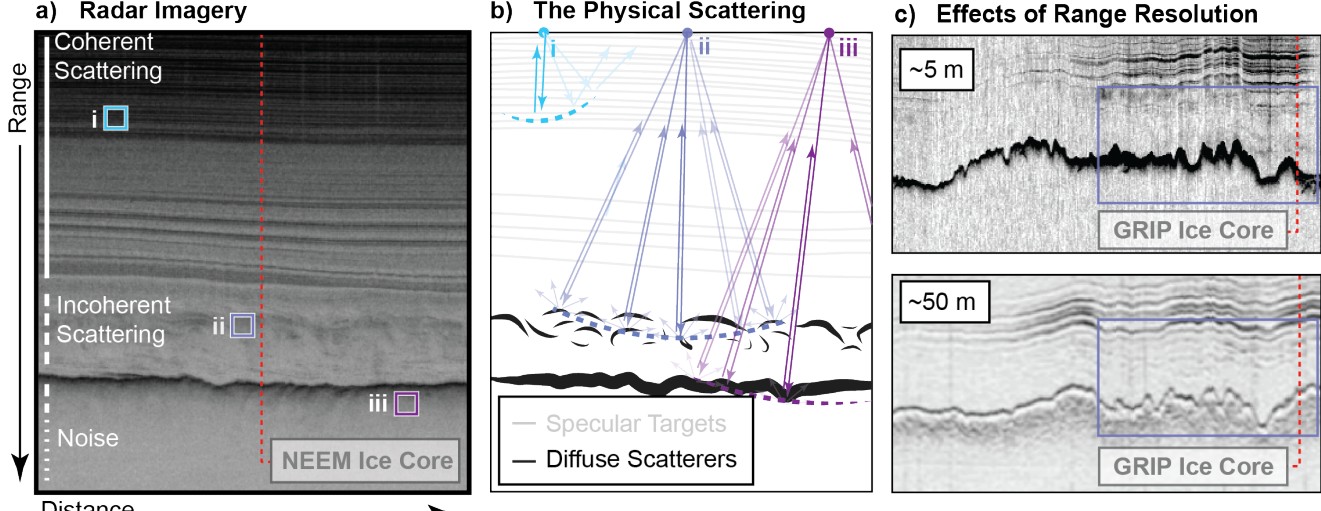

**Figure 2: (a)** Example radar image, **(b)** the ray-paths associated with scattering targets that contribute to individual pixels in the radar imagery, and **(c)** a pair of images highlighting the effect of system characteristics on the nature of deep scattering. Profiles presented in panel (c) were collected along sub-parallel tracks adjacent to the GRIP Ice Core site. Radar system characteristics for radargrams in (a) and (c) can be found in Supplementary Table 1.

To generate incoherent scattering, deep ice must differ from the planar, layered structure of the shallow ice column in some way. It may be that incoherent scattering occurs because chemical layering is mechanically disturbed in the deep ice and is no longer planar. Or, it may be that other processes (like dynamic recrystallization or grain rotation) acting locally (due to enhanced stress near obstacles to flow, transitions in the basal thermal state, or fluidity contrasts in the ice) introduce lateral heterogeneity in physical properties that produce incoherent scattering (Gerber et al., 2024). Here, we compile radar data from a variety of geophysical campaigns, including ground-based and airborne surveys conducted by the Center for Remote Sensing and Integrated Systems (CReSIS), the British Antarctic Survey (BAS), the University of Texas (UT), the University of Washington (UW), and the Alfred Wegener Institute (AWI) – (see Supplementary Table 1 for full system characteristics). From those data, we analyze representative, ice core adjacent radar images, and compare them to measurements of crystal orientation fabric and micro- and macro-scale structures, to test two hypotheses:

1.  That transitions in ice COF are collocated with (and likely induce) incoherent scattering.

2.  That small scale deformation of chemically distinct layering can induce incoherent scattering.

We are drawing from heterogeneous, historical data, which can make imagery intercomparison difficult. Because some systems used for site selection do not preserve phase information, we focus primarily on the amplitude and character of scattering, controlling for differences in system characteristics. Differences in image processing also have the potential to modify the expression and amplitude of incoherent scattering. Therefore, an important caveat of this work is that our interpretation of incoherent scattering only holds for imagery collected with radar hardware typical of the earth 2000's (with center frequences in the 100s of MHz) and the most common image post-processing (SAR focusing and along-track multilooking).

## 3. Data and Methods: Measurements Capturing the Fine- and Large-Scale Electrical Structure of Ice Cores

Folds and layer disturbances at all scales have been observed or inferred from ice core records in both Antarctica and Greenland. Some scales of folding are more easily detected – millimeter and centimeter scale folds can be measured directly within the 8-13 cm diameter ice cores. Folding at the 100s of meters scale is resolvable by radar. But all scales in between must be inferred using anomalous patterns of electrical conductivity, stable isotope or impurity concentrations, or physical and optical properties. We summarize the measurements that we use to identify deformation in deep ice below, and aim to relate radio-wave scattering phenomena to these observations.

Physical analysis of ice cores, including macro-scale visual observations and optical imaging (i.e. linescanners) (Faria et al., 2018; Jansen et al., 2015; Svensson, 2005), and alternating current and direct current electrical conductivity measurements (ECM) (Fudge et al., 2016; Wolff, 2000) provide the best direct measurement of small-scale features deep in the ice column. The resolution of typical linescan images is around 0.1 mm/pixel, allowing for observations of layers and their structure ranging from millimeter-scale undulations up to folds at the scale of the typical diameter of deep ice cores (Fig. 3). Data from ice core linescanning have shown wavy strata (e.g. WAIS -- West Antarctic Ice Sheet Divide (Fitzpatrick et al., 2014)), highly inclined strata (e.g. EDML -- EPICA Dronning Maud Land (Faria et al., 2018)), duplex and boudin-like structures (e.g. EastGRIP (Westhoff, 2021)), 10 cm-scale z-folds (e.g. NEEM -- North Greenland Eemian Ice Drilling (Jansen et al., 2015)), diffuse layering (e.g. NorthGRIP (Svensson et al., 2005)), and extreme growth of individual ice grains reaching diameters of up to 50 cm (e.g. EDML (Faria et al., 2018)), capturing unique forms of stratigraphic disturbance and in some cases, informing the depth associated with discontinuities in the climate record (Fig. 4).

To supplement imaging methods that capture small scale deformation, a range of chemical methods have been employed across deep ice core sites to identify major breaks in stratigraphic continuity and large-scale folding. Some breaks in continuity have been identified using chemical disagreement between ice cores. For cores in the same geographic region (e.g. GISP2 -- Greenland Ice Sheet Project Two, GRIP -- Greenland Ice Core Project, and NorthGRIP -- North Greenland Ice Core Project), divergence in electrical conductivity, $\delta^{18}O$ of ice ($\delta^{18}O_{ice}$), and impurity concentrations can be used to identify the onset of a discontinuous record (Johnsen et al., 2001). When looking across hemispheres, divergence in the profiles of globally well-mixed $\delta^{18}O$ of atmospheric $O_2$ ($\delta^{18}O_{atm}$) and $CH_4$ have been used to identify climate record discontinuity (Chappellaz et al., 1997; Landais et al., 2003). In cases where there are no cores that provide high resolution comparison, sudden shifts in the nature of the chemical signal (e.g. changes in chemical variability or abrupt changes in the gas-age ice-age difference, described as either the $\Delta$age between the ice and gas or the depth-shift separating gas and ice of a constant age) have been used to infer climate record discontinuities (Crotti et al., 2021; Dansgaard, 1982; Jouzel et al., 2007; Petit et al., 1999; Ruth et al., 2007). Chemical methods have also been used to reconstruct chronologies in heavily disturbed stratigraphy (Landais et al., 2003;

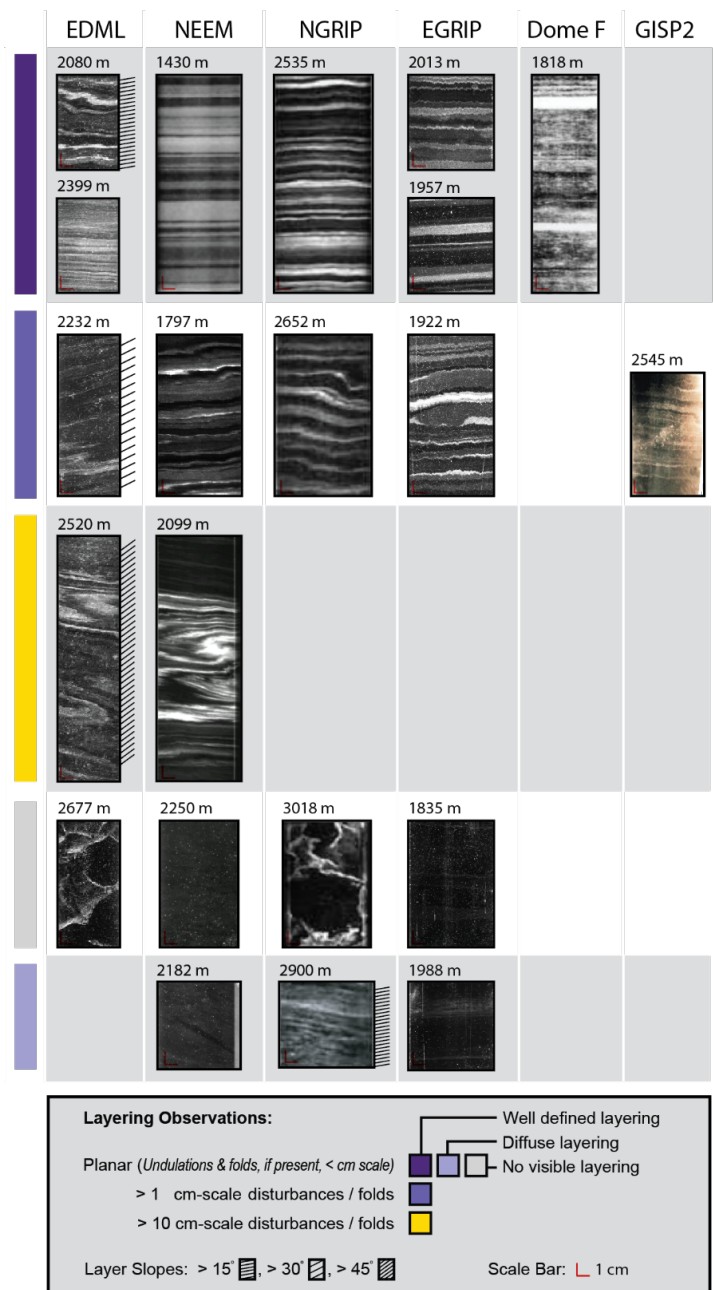

**Figure 3: Examples of linescan images capturing mm to >10-cm scale deformational structures. Microinclusion-rich ice strata scatter light creating bright horizons, or cloudy bands, revealing stratigraphic structure. Well-defined planar layering with mm-scale undulations is observed in all cores with available linescan images. Cm-scale deformational structures include z-folds (EDML and GISP2), cm-scale undulations (NEEM and NorthGRIP), and boudin-like structures (EastGRIP). Overturning folds that span over 10 cm of the ice column are observed at EDML and NEEM. Ice without layer structure can be due clear ice that lacks sufficient microinclusions for scattering (NEEM and EastGRIP) as well as ice with large individual crystal grains (EDML and NorthGRIP). Diffuse or weak layering is observed when microinclusions are minimal (NEEM and EastGRIP) or lacking clear layer structure (NorthGRIP). Linescan data is sourced from Faria et al., 2018 (EDML), Takata et al., 2004 (Dome Fuji), Kipfstuhl, 2009 (NEEM), Svensson, 2005 (NorthGRIP), Alley et al., 1997 (GISP2), Weikusat et al., 2020 (EastGRIP).**

NEEM Community Members, 2013; Raynaud et al., 2005; Souchez et al., 2002; Verbeke et al., 2002), and from those chronologies, identify overturned folding. These methods have in some places tentatively inferred (e.g., at Vostok and GRIP) and in other places clearly identified (at NEEM) folding on scales of 10-100 m.

In our analysis, we synthesize the literature on macro-scale stratigraphic disturbances, grouping and analyzing the effect of deformational structures on radar scattering based on reported fold size, slope inclination, and layer visibility. To do this, we identify the depth at which these features are observed (presented in Fig. 4) and compare the observed deformation patterns with collocated radar imagery. Most studies present examples of deformational feature types followed by qualitative descriptions of their frequency throughout the ice column; therefore, the reported ranges should be treated as zones of deformational structures with intermittent occurrence, rather than a continuous span of small-scale deformation.

In addition to measurements capturing the macro-scale, we present crystallographic analysis of glacial ice, typically performed using vertical and / or horizontal thin sections of ice cores. C-axis orientation can be measured with a range of techniques, including polarized light microscopy (Azuma et al., 1999; Weikusat et al., 2017; Wilson et al., 2003), x-ray diffraction and tomography (Miyamoto et al., 2011), sonic wave methods (Kluskiewicz et al., 2017), electron backscatter diffraction microscopy (Obbard and Baker, 2007), and open resonator methods (Saruya et al., 2024). Measurements of the bulk c-axis orientation of glacial ice gives us a direct constraint on how the polarization-dependent permittivity of ice might vary with depth, and therefore how variations in crystal orientation itself may be a source of scattering. C-axis measurements also provide information about the strain history of ice, with implications for larger-scale deformation in the ice column.

Historically, data from thin sections have provided the most robust evidence of differential strain at small scales, capturing fabric changes within a single 10 cm vertical thin section (e.g. NEEM (Montagnat et al., 2014)). But the logistics of thin section sampling limits their ability to capture some scales of vertical and horizontal variability in fabric. The distance between adjacent, discrete thin-section samples can be anywhere from 20 to 100+ m (e.g. EDML (Weikusat et al., 2013), Siple Dome (Gow and Meese, 2007), NorthGRIP (Wang et al., 2002), GRIP (Thorsteinsson et al., 1997)). New approaches to c-axis characterization may change what is possible in future studies of fabric derived scattering, as thick-section open resonator methods have been used to measure the clustering of crystal c-axes every 20 mm along the Dome Fuji core (Saruya et al., 2022, 2024). But for most available data, we are limited in our ability to quantitatively predict scattering from existing fabric measurements, as the magnitude of backscatter depends on the depth-rate-of-change of fabric. Instead, we focus primarily on qualitative comparison of fabric changes with radar images.

## 3. Results: Investigating the Sources of Incoherent Scattering

We present measured fabric and structural data together with radar imagery across 10 well sampled cores in Figure 4. We encourage readers to refer to Figure 4 often as we describe the relationships between structural data and the radiostratigraphy throughout section 3. In section 3.1, we evaluate the depth-agreement of scattering and known fabric transitions. In section 3.2, we evaluate the effect of small- and large-scale deformational structures on radar scattering. A full description of the ice core data used to generate Figure 4 can be found in Supplementary Table 2.

### 3.1 Crystal fabric transitions as a source of incoherent scattering

Given the enhanced stresses and therefore higher strain-rates near the base of ice sheets, one might expect monotonic but intensifying fabric development with depth. And at the majority of ice core drill sites, c-axis fabrics transition from a quasi-isotropic c-axis distribution at the top of the ice column to a strong single maximum lower in the column (e.g. Camp Century, Dye-3, GISP2, NEEM, EPICA Dome C (EDC), Talos Dome, GRIP), a product of the typical simple shear near the base of a glacier. Ice cores drilled at flank sites or otherwise away from ice divides often exhibit signs of uniaxial horizontal extension, and thus c-axis fabrics transition from quasi-isotropic to girdle-type fabric and then to a single maximum (e.g. NorthGRIP, Vostok, EDML). But variability in the impurity content (which changes with climate) can intensify fabric development and localize fabric transitions, with fabric strengthening typically coincident with higher impurity content (seen at Byrd (Faria et al., 2014), Camp Century (Faria et al., 2014), Talos Dome (Montagnat et al., 2012), EDC (Durand et al., 2009), NEEM (Montagnat et al., 2014), GISP2 (Gow et al., 1997), and Dye-3 (Langway et al., 1988)).

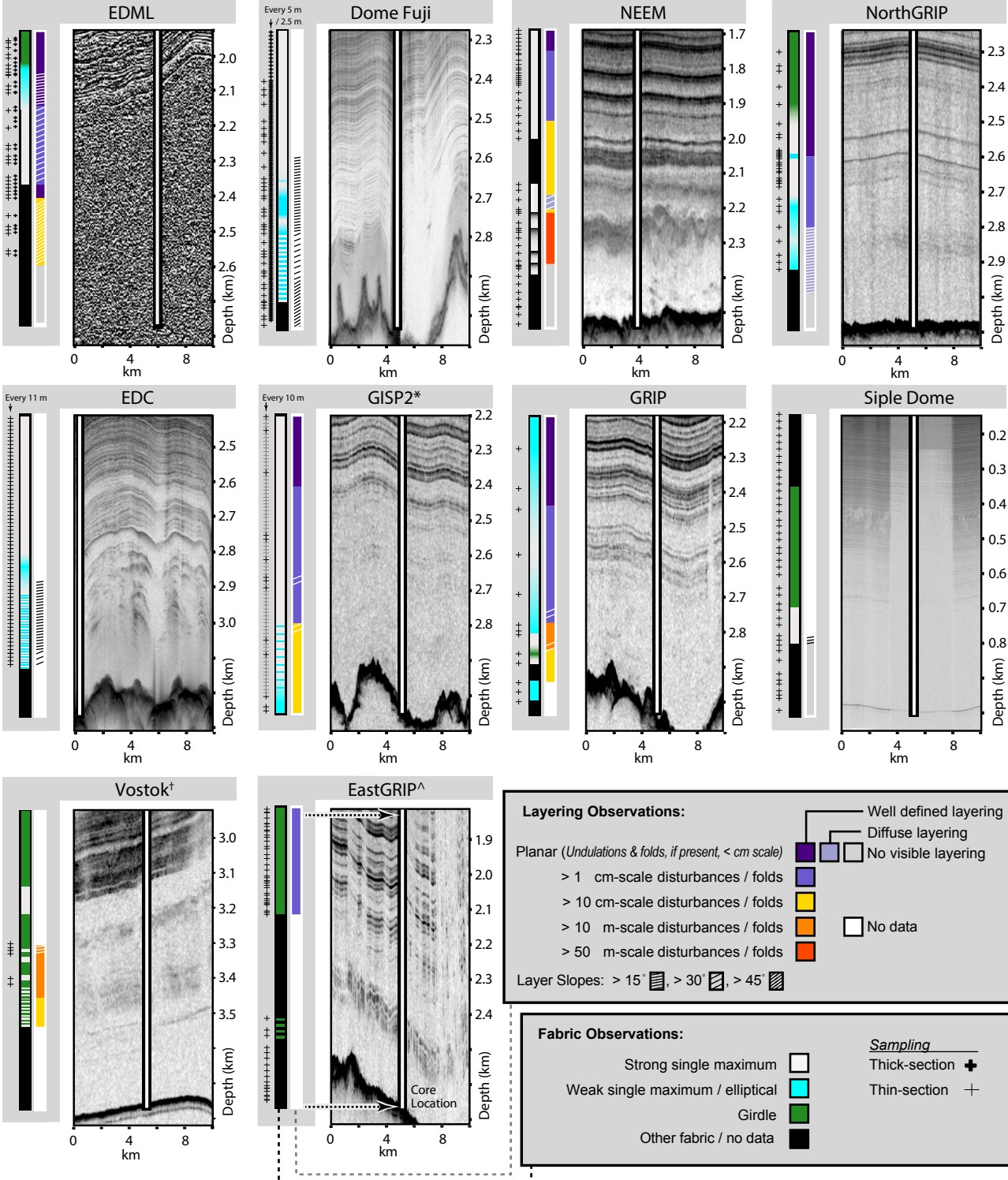

**Figure 4: Radargrams capturing deep ice at ice core drill sites with comprehensive fabric and stratigraphic deformation data. From left to right, each ice core panel presents a scatter plot marking sample depths where thin sections, and thick sections where applicable, were collected for crystal orientation fabric analysis, a colormap visualizing of fabric evolution with depth, a colormap visualizing layering evolution and layer slope observations with depth, and a 10 km length radar transect proximal to the ice core drill site. Fabric observations categorized as "other fabric" include multimaxima fabrics (e.g. at EastGRIP and NEEM). Radargrams span the bottom 850 m of each core and 50 m of bedrock. Backscatter power color scales are standardized to span 0.5% to 99% of the return power amplitude recorded in the presented depth range. Radar system characteristics can be found in Supplementary Table 1. The synthesized ice core data includes fabric observations: EDML (Eisen et al., 2007; Faria et al., 2018; Weikusat et al., 2013), Dome Fuji (Saruya et al., 2022, 2024), NEEM (Eichler, 2013; Montagnat et al., 2014), NorthGRIP (Wang et al., 2002), EDC (Durand et al., 2009), GISP2 (Gow et al., 1997), GRIP (Thorsteinsson et al., 1997), Siple Dome (Gow and Meese, 2007), Vostok (Obbard and Baker, 2007), EastGRIP (Stoll et al., 2024); and layering observations: EDML (Faria et al., 2010, 2018), Dome Fuji (Dome Fuji Ice Core Project Members, 2017), NEEM (Jansen et al., 2015), NorthGRIP (Svensson, 2005), EDC (Durand et al., 2009), GISP2 (Alley et al., 1995, 1997; Faria et al., 2014; Gow et al., 1997), GRIP (Alley et al., 1995; Dahl-Jensen et al., 1997; Johnsen et al., 1995; Landais et al., 2003), Siple Dome (Gow and Meese, 2007), Vostok (Lipenkov and Raynaud, 2015; Raynaud et al., 2005; Souchez et al., 2002), EastGRIP (Westhoff, 2021; Stoll et al., 2023). \*At GISP2, only some of the sampled thin sections have published data (indicated by the black + symbols), and †at Vostok, the original sampling rate is unpublished, with only a few thin sections and general observations available in the literature. ^At EastGRIP, visual characterization of cloudy bands combines folded features and weak layering into a single group (Stoll et al., 2023). We review the published linescan images at EastGRIP and present approximate depths of these two types of layering in Fig. S1.**

Abrupt fabric transitions occur within most ice cores in Greenland (e.g. Camp Century, Dye-3, GISP2, and NEEM), where a significant change in impurity deposition at the Holocene-Wisconsin climate transition drives an abrupt strengthening or transition to a vertical-maximum fabric (Faria et al., 2014). In some places, we see a co-located scattering horizon associated with these abrupt transitions in fabric. At NEEM, a transition from a weak vertical girdle to strong single maximum fabric occurs at 1419 m and is coincident with a diffuse reflector in the radargram (Fig. S2). Similar reflectors appear at isolated fabric transitions in Antarctica as well. At Siple Dome, the c-axis fabric transitions from a vertical girdle to a single maximum at 700 m, with a corresponding diffuse reflector in the radar data. At EDML, the c-axis fabric transition from a vertical girdle to a strong single maximum between 2025 m and 2045 m has been identified as the origin of the reflector at 2035 m (Eisen et al., 2007). These reflectors appear less specular (with trailing energy after the initial arrival) than other isochronous layering within radar imagery.

Where we see well sampled gradual transitions in fabric (spanning 50-100 m of the ice column) we observe both diffuse bands of incoherent scattering as well as laterally heterogeneous incoherent scattering. At EDC, the strong single maximum fabric at 2800 m gradually transitions to a broad single maximum fabric at 2857 m and returns to a strong single maximum fabric at 2900 m (Durand et al., 2009). This fabric transition is roughly coincident with the transition from coherent isochronal strata to a single diffuse incoherent scattering layer observed around 2825 m. At Dome Fuji, the strong single maximum fabric at 2660 m gradually weakens before returning to a strong single maximum fabric again at 2760 m (Saruya et al., 2024). This fabric transition appears roughly coincident with a weak diffuse incoherent scattering layer observed at ~2700 m in the radargram (Fig. S3.a).

In many places, especially where annual layer thickness is compressed significantly at the base of the ice column, alternating fabrics have been observed. At Vostok, from 2700 to 3315 m depth, the core alternates between coarse-grained ice with girdle-type fabric and fine-grained ice with single-maximum fabric every ~100 m (Obbard and Baker, 2007). Within the girdle-type fabric zone between ~3220 and 3315 m, we see weakly banded incoherent scattering (3220 – 3290 m). Between ~3315 and 3450 m, alternations between girdle-type and single-maximum fabric occur approximately every ~20 m (Lipenkov and Raynaud, 2015). This zone of increased fabric alternation overlaps with both the no echo zone between ~ 3290 and 3360 m and the upper depths of a weakly banded incoherent scattering unit (~3360 – 3490 m) in the radargram. At GRIP, each of the five thin sections sampled between 2800 and 2950 m depth show alternating fabrics. At GISP2, coarse-grained layers with fabrics that deviate from the strong single maximum are observed at increasing frequencies below 2800 m (Gow et al., 1997). While interpretation of the GISP2 and GRIP radargrams is challenging below 2800 m, 35 km length radar transects show laterally heterogeneous incoherent scattering in that depth range (Fig. S4).

While it is challenging to describe fabric variability at all scales from thin-sections due to their irregular sampling frequency, the smallest scale of fabric variability has been observed or inferred at centimeter-scales, including at Vostok, EDC, Dome Fuji, and EastGRIP.

- At Vostok, fabric alternations occur at cm-scale wavelengths from 3450 m until the transition from meteoric to accreted ice at 3538 m (Lipenkov and Raynaud, 2015). This overlaps with an echo-free zone in the radargram.
- At EDC, ice below 2800 m consists of alternating layers with high impurity content (consistently presenting strong single maximum fabric) and layers with low impurity content (with an associated broad single maximum fabric). After the gradual transition into and out of a broad single maximum fabric at 2850 m, fabric transitions below 2920 m become more local. High spatial sampling (every 0.5 m) between 2933 and 2955 m revealed fabric alternations between each sample (Durand et al., 2009). Unlike at Vostok where the onset of rapid fabric transitions coincides with the start of the echo free zone, the onset of rapid fabric transitions at EDC is associated with thick and sometimes discontinuous bands of incoherent scattering (2900 – 3050 m) in the radargram.
- At Dome Fuji, cm-scale fluctuations from the single maximum fabric, observed by increases in the standard deviation of $\Delta\varepsilon$ (the difference in the relative permittivity, $\varepsilon$, between vertical and horizontal planes), begin around 2400 m and intensify through the base of the ice column (Saruya et al., 2024). The increase in fabric fluctuations between 2400 and 2650 m has no obvious effect on the coherent continuous layering observed in the radargram. However, the Dome Fuji radargram transitions to a zone of laterally homogenous incoherent scattering at 2900 m. Notably, the precise depth of that transition is difficult to constrain in the radar image, due to the combination of increasing layer inclinations (Dome Fuji Ice Core Project Members, 2017) and strong scattering from borehole fluid in the ice core cavity (Fig. S3).
- At EastGRIP, rapid transitions between vertical girdle and multi-maximum fabrics are observed between 2417 and 2484 m, with a strong multi-maximum fabric established below 2500 m (Stoll et al., 2024). The depth range of the

rapid fabric transitions coincides with a layer-conformal package of incoherent scattering. Banding within the package
of incoherent scattering is not layer-conformal, and the bands are defined by laterally traceable, abrupt drops in power
with depth (rather than laterally traceable, abrupt increases in returned power as we see in the coherent layering
above). We describe these traceable lows in power as "nulls", likely the product of destructive interference in scattered
energy returning to the radar from multiple directions. The expression of the nulls in the imagery is polarization
dependent (Fig. S1; Nymand, 2024 Fig. 3.5) suggesting that this entire scattering package is a result of the fabric.

At NEEM, four sequences of abrupt and then gradual fabric transitions are linked to large-scale deformation starting at ~2200
m. In this section of the ice core, the same oxygen isotope sequence (and its associated fabric gradient, from multi-maxima
fabric to single maximum fabric) is repeated, with abrupt fabric transitions at the boundaries between sequences. This is
attributed to overturned folds at the base of the ice column, in part, facilitated by rheologic differences in the ice that also
produce the abrupt fabric transitions. At these depths we see strong incoherent scattering that is highly laterally variable. Here,
both fabric and larger-scale deformation likely play a significant role in the nature of the scattering, with folding introducing
lateral heterogeneity in material properties that has not been identified at other ice core sites.
**3.2 Folding as a source of incoherent scattering**
Millimeter-scale disturbances are likely present in most deep glacial ice, given their ubiquity in ice cores. But we find little
evidence that deformation at that scale impacts the radiostratigraphy directly. In the South Pole Ice Core (SPICEcore), inclined
and pinched cloudy bands are observed starting at 1000 m and continue intermittently through the end of the core (Fegyveresi
and Alley, 2018) without any noticeable impact on radar scattering. Crystal striping at GISP2 is observed starting at 2200 m,
coincident with the onset of small-scale undulations in linescan images (Alley et al., 1997). But similar to SPICEcore, there is
no associated change in the nature of radar layering. Millimeter-scale z-folds at GRIP first appear at 2438 m and at 2437 m at
GISP2 (Alley et al., 1997), which does coincide with a drop in power of coherent scattering layers.  But there is a commensurate
drop in the ice conductivity variability associated with changes in dust deposition, which better explains that change. Thus, we
rule out millimeter-scale folding as a significant contributor to the radar signal observed at these locations.

Stratigraphic disturbances at the centimeter-scale are apparent in all cores with available data. In previous work, this scale of
deformation has been invoked as a mechanism for the "echo free zone", with the idea that folding effectively homogenizes
dielectric contrasts at the scale of the resolution of the radar (Winter et al., 2017). At EDML and WAIS Divide, the onset of
cm-scale disturbance does appear to be collocated with the apparent echo free zone. In both radar images, however, there is a
gradual diminution of returned power with depth. It is possible that measured disturbances do reduce the intensity of back-
scatter without eliminating it entirely. But there is laterally-continuous layering (with strong back-scatter intensities) in regions
of cm-scale disturbances at NorthGRIP, NEEM, EastGRIP, and GRIP, and in regions with disturbances at the scale of 10 cm
at NEEM. Radar data at NEEM show no change in scattering behavior associated with deformation at this scale. This seems
to imply that these radar systems (with range-resolutions of 2.8 m to 5 m (Supplementary Table 1)) are insensitive to
deformation at this scale.

Larger scale folding does seem to have an effect on the radiostratigraphy. Deeper in the NEEM core, where chemical analyses
reveal six zones of disturbed ice, including two large 50 and 100 m thick folded layers of inverted early glacial ice (NEEM
Community Members, 2013), high amplitude but laterally variable incoherent scattering can be seen in the radar imagery.
Deformation at this scale, thought to be in part due to rheological differences between the glacial and interglacial ice (NEEM
Community Members, 2013), is coincident with a loss of coherent banding in the linescan imagery and an increase in the
lateral heterogeneity of intensity in incoherent backscatter. Above 3460 m depth at Vostok, folding is also inferred at the meter
scale and larger (Lipenkov and Raynaud, 2015). Similarly, there is incoherent scattering in the image at these depths, although
the amplitude of the backscatter is weaker, and lateral heterogeneity less pronounced. Finally, at GRIP, tentative chronological
reconstructions of disturbed ice below 2750 m show significant disruption and folding on the scale of 10s of meters between
2780 to 2850 m. And while near the ice core, this depth-range corresponds with a unit of weak incoherent scattering, at the
10s of kilometers scale, there is significant variability in the amplitude (Fig. S4).
**4. Discussion: Using Incoherent Scattering in Ice Core Site Selection**
There is compelling evidence that incoherent scattering can arise from fabric transitions in the deep ice, and the quality of that
scattering could be diagnostic of large-scale deformation that is co-located with the smaller-scale fabric development. If true,
then incoherent scattering might be used to improve ice core site selection. We test that theory at 16 ice core sites, by first
subdividing core-adjacent radar imagery into five types of signal (Figs. 5a and 5b):

347        1.   Laterally continuous coherent scattering (that is, clear isochronal layering)

348        2.   Diffuse but banded scattering

349        3.   Laterally homogenous incoherent scattering

350        4.   Laterally heterogeneous incoherent scattering

351        5.   No signal (or rather, signal levels at or below the noise floor of the system).


We then compare these scattering types to known breaks in the continuity of the associated ice cores (see Appendix A for the
observational basis for each labelled break).

Across these core sites, continuous coherent scattering is almost exclusively found above known breaks in the climate record.
This type of scattering appears below the break in a climate record in only one ice core, and that is Vostok, where the interface

between accreted and meteoric ice and a layer of mineral inclusions from the lake bed (Turkeev et al., 2021) define two clear reflection horizons. As a result, in typical glaciological environments, continuous coherent scattering is a robust indicator of ice core continuity. At the studied core sites, where diffuse but banded scattering sits immediately below laterally continuous layering (as is the case at EDC and EastGRIP), there are no associated breaks in measured climate records. This supports the idea that banded but incoherent scattering is not an indication of disturbed basal ice.

Where we see laterally homogenous incoherent scattering, as in Camp Century, EDC, Dome Fuji, and NorthGRIP, it occurs within sections of ice with a continuous climate record. This likely indicates fabric transitions that are themselves defined weakly by depositional impurities, and thus, the shape of the scattering band is roughly parallel to the isochronous layering. At Vostok, we see incoherent scattering that is laterally heterogeneous in its intensity but is otherwise layering conformal, directly above and ~100 - 200 m below the broken climate record. These two bands of incoherent scattering are qualitatively indistinguishable, and demonstrate the challenge of interpreting the quality of the climate record within regions characterized by bed conformal laterally heterogeneous incoherent scattering.

But where we see laterally heterogeneous incoherent scattering that is layering non-conformal (as in GISP2, GRIP, and NEEM) it occurs below breaks in the continuity of the observed climate record. We show that the source of the backscattering is transition in the crystal fabric of the glacier, and its macro-scale expression comes from the nature of the vertical and lateral heterogeneity in fabric. In those places, it is possible that the same ice rheology contrast that facilitated a fabric transition interacts with the complex, local, basal stress regime to enable multi-meter scale deformation. This induces lateral variability in the backscatter intensity, and can be taken as a significant risk for a disturbed climate record.

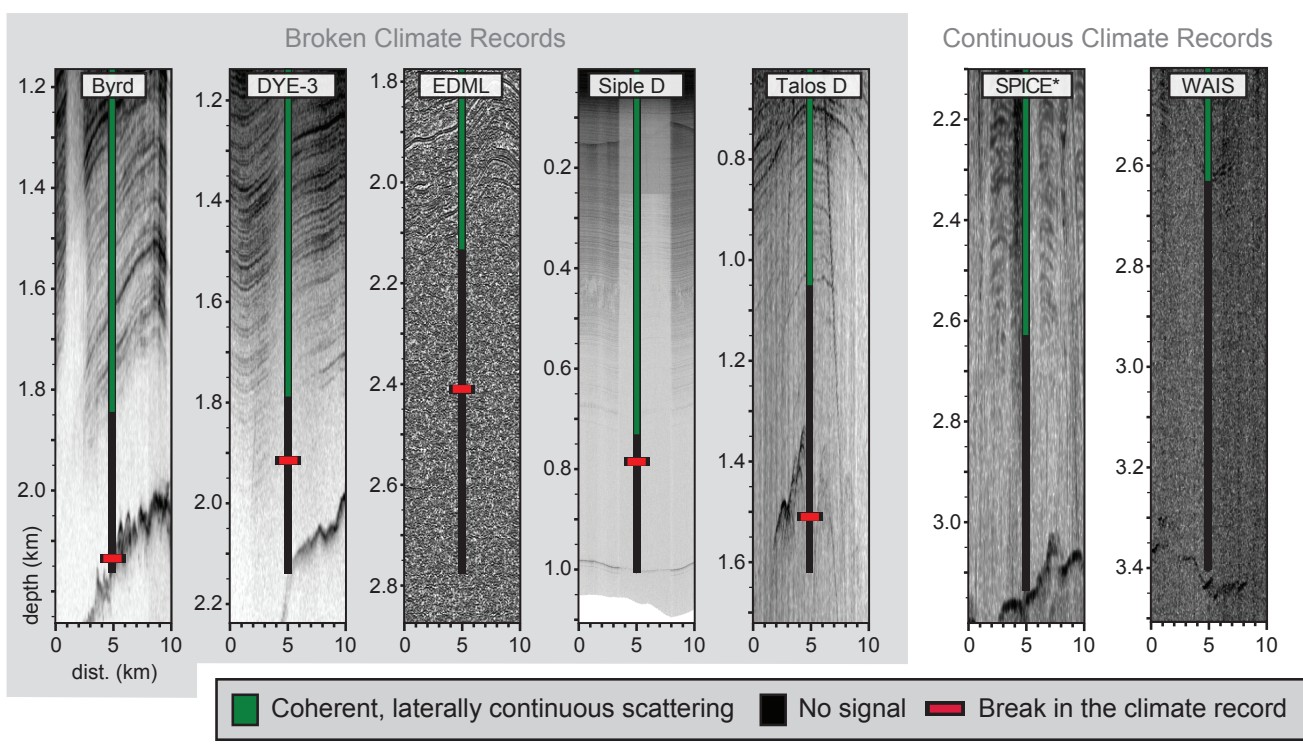

Coherent, laterally continuous scattering    No signal    Break in the climate record

*SPICECore drilling ceased at 1500 m. The continuity of the climate record below 1500 m is unknown.

**Figure 5a**

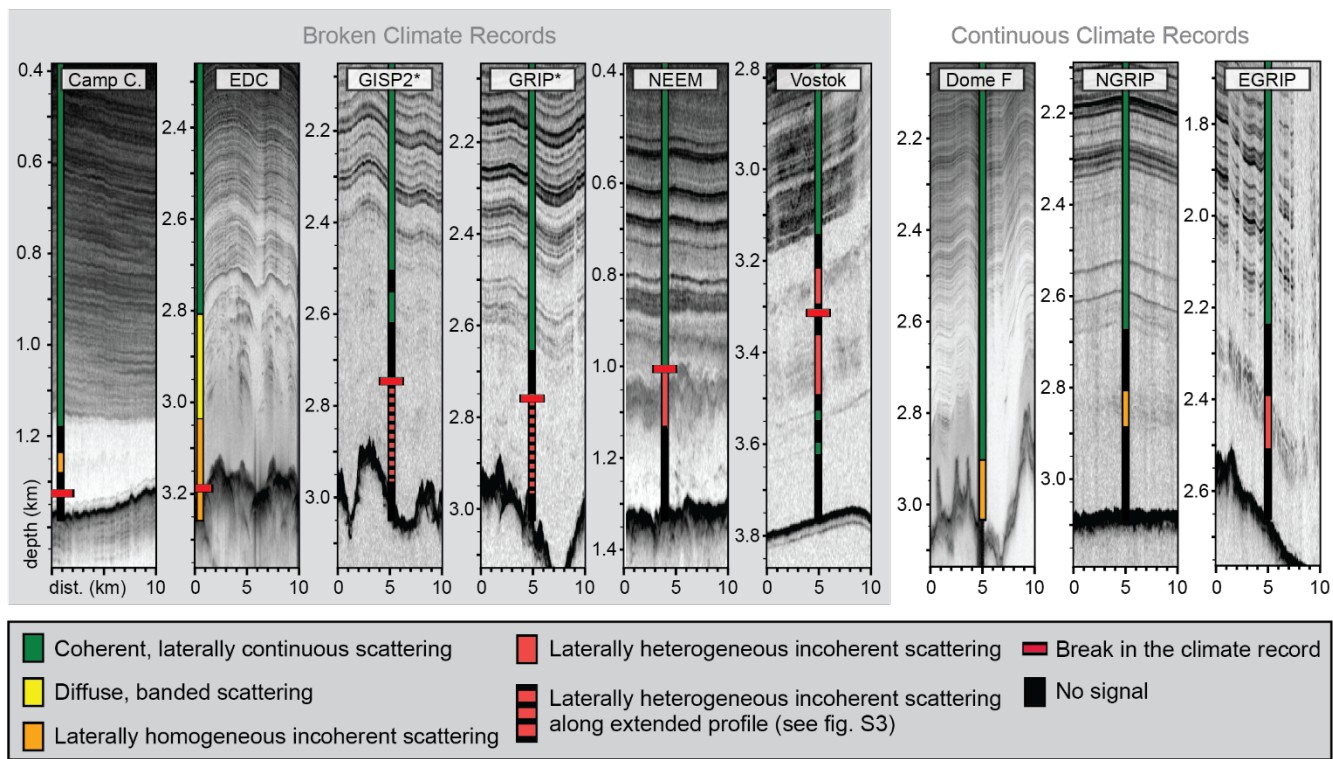

**Figure 5b**

**Figure 5: 10 km length radar profiles collected proximal to the 16 ice core drill sites. Radargrams are 1100 m in depth, spanning the bottom 1000 m of each ice core. Backscatter power color scales are standardized to span 0.5% to 99% of the return power amplitude recorded in the presented depth range. Radar system characteristics can be found in Supplementary Table 1. The depth of the broken climate record, described in Appendix A, is marked at the relevant core sites. The quality of radar scattering at the ice core drill site is color-coded based on categorization as coherent, diffuse and banded, incoherent and laterally homogeneous, incoherent and laterally heterogeneous, or no signal (below the noise floor). (a) Radargrams from Byrd, Dye-3, EDML, Siple Dome, Talos Dome, South Pole, and WAIS Divide exhibit coherent laterally continuous scattering until the noise floor of the radar instrument is reached. Lack of scattering once the instrument reaches the noise floor inhibits interpretation of the quality of the climate record at depth. (b) Radargrams from Camp Century, EDC, GISP2, GRIP, NEEM, Vostok, Dome Fuji, NorthGRIP, and EastGRIP exhibit a variety of incoherent scattering patterns. Incoherent scattering is observed within both continuous climate records at Camp Century, EDC, Vostok, Dome Fuji, NorthGRIP, and EastGRIP, and broken climate records at EDC, GRIP, GISP2, NEEM, Vostok. \*Laterally heterogeneous incoherent scattering at GISP2 and GRIP is best observed along the extended 35 km radar transects in Fig. S4.**

## 5. Conclusions

Based on comparison between ice core data and ice-penetrating radar imagery at ice core sites, we show that diffuse and incoherent scattering is often collocated with transitions in the crystal orientation fabric of the ice. Transitions in fabric are a product of the local stress regime, but they are localized by differences in grain size. High concentrations of impurities tend to reduce local grain-size and enhance deformation rates, so where climatically driven variations in impurities change the strength of the ice, one might also expect more abrupt contrasts in fabric that back-scatter radio waves. In this way, fabric controlled

scattering may be roughly isochronous, although we show that fabric interfaces do not manifest as abrupt, specular reflectors the way chemically induced layering does in radar imagery.

In the deep ice, where stresses are high, the age-depth scale is compressed, and global changes in impurity deposition are expressed over narrower depth ranges, we might expect fabric induced scattering to be common. The nature of the fabric transition, and the spatial heterogeneity in the transition, define whether or not the scattering will appear as coherent layering, a diffuse scattering horizon, laterally homogenous incoherent scattering, or laterally heterogeneous incoherent scattering. In addition, ice fluidity contrasts at fabric boundaries facilitate small- and large-scale folding. At small scales (below ~1 m), folding seems to have little impact on existing radar data. But large-scale folding, where present, results in complex scattering targets in the subsurface, and induces significant lateral heterogeneity in the incoherent scattering intensity and complex scattering horizons. Where this is observed at existing ice core sites, it seems indicative of discontinuities in the ice core climate record.

A final consideration when thinking about fabric induced incoherent scattering is the relationship between permittivity contrasts (as experienced by the propagating radio-wave) and radio-wave polarization. For fabric intensification (for example, a weak single maximum to a strong single maximum fabric) there will be a change in permittivity for all radar polarizations, and scattering will likely appear isotropic. For fabric transitions (for example, from a girdle to a single maximum fabric) it is possible for some polarizations to exhibit scattering and others to have low backscatter or apparent echo free zones. This anisotropic character merits further study at places like Siple Dome, EDML, EastGRIP, and Vostok, where girdles are seen in the deep ice.

As is true for discussions of the "echo free zone", we show that conversations about the "basal layer" observed in Greenland and Antarctica must start from the understanding that deep scattering (or its absence) depends on system characteristics and physical properties of the ice. Using only amplitude information to diagnose the source of scattering is therefore inherently limited, not just by the non-unique nature of geophysical imaging (both echo free zones and deep incoherent scattering could arise from multiple mechanisms) but also due to subjective choices made during image processing. Future surveys with phase-coherent data should augment amplitude analysis with along-track direction-of-arrival analysis to get a quantitative measure of specularity (as in Heister and Scheiber, 2018). But from the historical data, we show that a common mechanism for incoherent scattering in deep ice is transition in ice crystal fabric. We find that qualitative differences in the nature of incoherent scattering can aid in evaluating the suitability of future ice core sites. But most importantly, we hope to emphasize that incoherent scattering is signal, not noise, and more work should be done to better interpret this often overlooked component of radar imagery.

## 6. Data Availability

The radar data and associated metadata used in this analysis is available in the accompanying data dictionary (https://doi.org/10.7910/DVN/JAQJWZ).

## 7. Author Contribution

EM synthesized data from the literature on physical and chemical properties of ice cores and identified radar data from CReSIS, BAS, UT, UW, and AWI. All authors contributed to study design, radargram interpretation, figure creation, and writing of manuscript.

## 8. Competing Interests

The authors declare that they have no conflict of interest.

## 9. Acknowledgements

This work was funded through the Center for Oldest Ice Exploration (NSF-2019719). It also represents an aggregation of a tremendous amount of work from previous scholars studying ice cores, and we would like to thank those communities and encourage suggestions from those scholars for new ways to connect ice penetrating radar to measurable ice core quantities.

## Appendix A. Known Layer Disturbances and Ice Core Continuity Problems

Of the cores studied, 6 show only minor signs of layer disturbances, and contain a continuous climate record through the full depth range of the ice core. Those are EastGRIP, Dome Fuji, NorthGRIP, SPICEcore, and WAIS Divide. Of the other 10 cores, 5 have well identified breaks in their climate record, and 4 are likely discontinuous (although the exact stratigraphic break is not well identified), and 1 has conflicting observations of discontinuity. A full list of the ice core data used for these observations, including the oldest age of the continuous climate record, can be found in Fig. S5 and Supplementary Table 2. Here, we describe the observational basis for claims of both continuous and broken climate records.

### A.1 Cores with Clear Evidence of Stratigraphic Discontinuities

*(Alphabetically: EDML, GRIP, GISP2, NEEM, Talos Dome, Vostok)*

**EPICA (European Project for Ice Coring in Antarctica) Dronning Maud Land, EDML (Length: 2774 m | Break: 2417 m | Percentage Disturbed: 12.9%):** The chronology called EDML1 has been established for the top 2417 m of the EDML ice core. The top 2366 m of the core is matched to the EDC3 chronology using volcanic signatures (dielectric profiling (DEP), $SO_4$ concentrations, and electrolyte conductivity measurements) (Ruth et al., 2007). Three tie points between the EDC3 chronology and EDML core are matched between 2366 and 2415 m using insoluble dust concentrations, $\delta^{18}O$, and $\delta D$, however these matches are considered uncertain with estimated errors up to several thousand years (Ruth et al., 2007). Macrostructure analysis of linescan images between 2400 and 2500 m shows evidence of large-scale folding (Faria et al., 2010).

**Greenland Ice Core Project, GRIP (Length: 3029 m | Break: ~2750 m | Percentage Disturbed: 9.2%) and Greenland Ice Sheet Project Two, GISP2D (Length: 3053.4 m | Break: ~2750 m | Percentage Disturbed: 9.9%):** $CH_4$ and $\delta^{18}O_{atm}$ data from both GRIP and GISP2 show evidence of stratigraphic disturbance in the bottom 10% the ice cores. Above 2750 m $CH_4$ and $\delta^{18}O_{atm}$ values vary synchronously between GRIP and GISP2, but below 2750 m, the chemical profiles diverge, showing large and significant fluctuations which are not present in the undisturbed ice from the Vostok 3G core (Chappellaz et al., 1997).

**North Greenland Eemian Ice Drilling, NEEM (Length: 2540 m | Break: 2209.6 m | Percentage Disturbed: 13%):** At NEEM, an abrupt discontinuity in the $\delta^{18}O_{ice}$ at 2209.6 m marks the end of synchronization with the NorthGRIP GICC05 extended timescale. Additional discontinuities in the $\delta^{18}O_{ice}$ subdivide the bottom 13% of the core into six zones of disturbed stratigraphy. These correspond with similar shifts in other atmospheric gas measurements ($CH_4$, $\delta^{18}O_{atm}$, $N_2O$, $\delta^{15}N$ of $N_2$). Within the upper five zones, the layering is thought to be unbroken (based on continuous records of $N_2O$, $\delta^{15}N$ of $N_2$, dust, or electrical properties), with timescales for each of the upper five zones reconstructed by synchronizing NEEM $\delta^{18}O_{atm}$ and $CH_4$

profiles with NorthGRIP and EDML records. The timescales for these zones include inverted, mirrored, and folded ice up to
100 m thick (NEEM Community Members, 2013).

**TALos Dome Ice CorE, TALDICE (Length: 1620 m | Break: 1548 m | Percentage Disturbed: 4.4%):** At Talos Dome,
Crotti et al. identify a break in stratigraphic continuity at 1548 m using analysis of $\delta^{18}O_{atm}$, $\delta D$, and 81Kr dating, described
below (Crotti et al., 2021). TALDICE $\delta^{18}O_{atm}$ and $\delta D$ measurements were matched to the EDC $\delta^{18}O_{atm}$ and $\delta D$ record through
visual synchronization through 1548 m depth. Below 1548 m, the amplitude of $\delta^{18}O_{atm}$ fluctuations is damped, making
synchronization with the EDC record uncertain. Similarly, below 1548 m, the TALDICE $\delta D$ signal becomes asynchronous
with the EDC record. $^{81}$Kr dating of three samples below 1548 m depth revealed that ice from 1613 - 1618 m had comparable
age to samples from 1559 - 1563 m and 1573 - 1578 m depth, indicating a disturbed age-depth relationship.

**Vostok 5G-5 (Length: 3658 m | Break: 3311 m | Percentage Disturbed: 9.5%):** The stratigraphy in the bottom 9% of the
Vostok 5G core is divided between 228 m of disturbed meteoric ice, and 119 m of accreted lake ice. In the upper part of the
disturbed meteoric ice, the lack of depth-shift between $\delta D_{ice}$ and gas measurements ($CO_2$ and $CH_4$) is interpreted by Souchez
et al. as evidence of folding and intermixing (Souchez et al., 2002). Observations of ash layers with depth-varying inclinations
supports interpretation of large-scale folding. In the lower part of the disturbed meteoric ice, damped variation of $\delta D_{ice}$ and
trace impurity distributions (Na+, Cl-, non-sea salt Mg++ and Ca++), physical observations of interbedded fine-grained
(presumably glacial) and coarse-grained (presumably interglacial) ice, and the presence of bed material in the bottom 100 m
of the disturbed meteoric ice, is interpreted as further evidence for stratigraphic deformation (Lipenkov and Raynaud, 2015;
Souchez et al., 2002). At 3538 m depth, the transition between meteoric and accreted ice is apparent from the $\delta D_{ice}$/ $\delta^{18}O$
fingerprint of freezing processes (Jouzel et al., 1999). At this depth, sudden transitions to lower total gas content, increased
crystal size, low ECM values, increased $\delta D_{ice}$, and decreased deuterium excess, provide further evidence for the
meteoric/accreted ice transition (Jouzel et al., 1999).
**A.2 Cores that Likely Contain Stratigraphic Discontinuities or Conflicting Observations of Discontinuity**
*(Alphabetically: Byrd, Camp Century, EPICA Dome C, Dye-3, Siple Dome)*

**Byrd Station '68, BYRD 68 (Length: 2164 m | Break: 2135-2144 m | Percentage Disturbed: ~1%):** A chronology for the
upper ~99% (2144 m) of the Byrd core has been established by synchronizing Byrd, GRIP, and GISP2 $CH_4$ profiles (Blunier
and Brook, 2001). Gas volume measurements from the bottom 10 m of the core (2154 - 2164 m) suddenly approach zero at
4.83 m above the bed, revealing the transition between meteoric ice and accreted subglacial meltwater (Gow et al., 1979). The
bottom 4.83 m of non-meteoric ice contain horizontal bands of basal debris including sand, clay, and pebbles as large as 8 cm
in diameter (Gow et al., 1979). Grootes et al. 2001 observe that the Byrd $\delta^{18}O$ record becomes asynchronous with Taylor Dome
and Vostok record around 2135 m.

**Camp Century, CC 63-66 (Length: 1387.4 m | Break: ~1350 m | Percentage Disturbed: 2.7%):** The integrity of the Camp Century climate record is uncertain below 1310 m depth where $\delta^{18}O$ profiles of Camp Century, GRIP, and GISP2 become asynchronous (Johnsen et al., 2001). Correlation of a smoothed Camp Century $\delta^{18}O$ profile with benthic foraminifera record from deep sea core RC11-120 provides a tentative extension of the chronology through about 1330 m, the depth of the inflection point associated with Marine Isotope Stage (MIS) 5d (Dansgaard et al., 1985). A dramatic cold event at 1340 m is associated with a similar $\delta^{18}O$ fluctuation in the disturbed section of the GRIP core at 2800 m (Johnsen et al., 2001). Johnsen et al. describe dramatic fluctuations in $\delta^{18}O$ below Greenland Interstadial (GI) 23 in the GRIP, GISP2, and Camp Century cores which are not represented in the continuous $\delta^{18}O$ signal from Vostok (Chappellaz et al., 1997).

**EPICA Dome C, EDC99 (Length: 3260 m | Potential Break: ~3200 m | Percentage Disturbed: ~1.8%):** The continuity of the upper 98% (3200 m) of the EDC99 core is evidenced primarily through matching of $\delta D_{ice}$ to the deep-sea benthic $\delta^{18}O$ record (Jouzel et al., 2007). Additional matching of enhanced $^{10}Be$ deposition to Matuyama-Brunhes geomagnetic reversal between 3160 and 3170 m (Jouzel et al., 2007) and matching of $CO_2$ and $CH_4$ profiles to MIS18 and 19 between 3160 and 3185 m further support the continuity of the upper 98% of the core. Below 3200 m, there is contradictory evidence about the continuity of the climate record. Measurements of $\delta D$, total air content, gas composition, and dust content suggest continuity to bedrock, while $\delta^{18}O_{atm}$, visible inclusions, length of the glacial period, and variability of chemical species distribution suggest altered stratigraphy (Tison et al., 2015).

**DYE-3, DYE3 79-81 (Length: 2037 m | Break: 1940 m | Percentage Disturbed: 4.8%):** At DYE-3, the continuity of the climate signal is lost between 1900 and 1987 m. Initially, Dansgaard et al. 1982 correlated fluctuations between the $\delta^{18}O$ measurements at DYE-3 and Camp Century through 1987 m depth. Between 1987 and 2010 m, DYE-3 $\delta^{18}O$ values are quasi-constant, and interpreted as evidence of folded layers. Later, comparison of the $\delta^{18}O$ values between DYE-3 and GRIP led Johnsen et al., 2001 to identify Greenland Interstadial (GI) 8 at 1900 m as the last undisturbed match point between the two records. However, Johnsen et al. would still identify two match points in the deeper ice: GI 12 (~1925 m) and GI 14 (~1940 m). Recent analysis of $\delta^{15}N-N_2$ and $CH_4$ gas records may suggest stratigraphic disturbance beginning at 1895 m depth (Buizert et al., 2024). Since the scale of the gas record disturbances has not yet been quantified, in our analysis we have used 1940 m as the depth of the broken climate record. $CO_2$ and $CH_4$ measurements of the bottom 27 m of silty ice have been used to identify 4 distinct zones of highly deformed basal ice (Verbeke et al., 2002).

**Siple Dome A, SDMA (Length: 1004 m | Break: ~800 m | Percentage Disturbed: ~20%):** The integrity of the Siple Dome climate record is uncertain in the bottom 200 m of the core, however a precise onset depth for the disturbed ice is poorly constrained. A chronology for the 514 - 854 m section of the core was established by synchronizing Siple Dome, GISP2, and GRIP $CH_4$ profiles (Brook et al., 2005). Below 854 m, the methane data becomes sparse however a possible chronology has

been proposed between 854 and 920 m based on the matching of a single inflection point in the $\delta^{18}O_{atm}$ profile of Siple Dome core at 920 m with a corresponding GISP2 $\delta^{18}O_{atm}$ inflection point (Brook et al., 2005). Macro and micro-scale physical observations by Gow and Meese suggest an interrupted climate record by 800 m depth, summarized here (Gow and Meese, 2007). Between 560 and 800 m, sequences of inclined layering occasionally surpassing 10 degrees as well as reversed dips are observed. Below 800 m the core is highly fractured, limiting any further observations of layer structure. Around 700 m, the c-axis fabric shifts suddenly to a single maximum corresponding to a stress regime dominated by strong horizontal shear. Around 800 m, the c-axis fabric shifts back to a multi-maxima fabric.

**A.3 Cores with No Significant Break in Continuity**

*(Alphabetically: EastGRIP, Dome Fuji, NorthGRIP, SPICEcore, WAIS Divide)*

**East Greenland Ice Core Project, EastGRIP (Length: 2663 m):** Initial assessment of the continuity of the EastGRIP climate record has been performed through synchronization of DEPs and ECMs to NEEM and NGRIP datasets. These techniques have been used to establish the GICC05-EGRIP-1 timescale for the upper 1383.4 m of the core (Mojtabavi et al., 2020). Preliminary comparison of EastGRIP and NGRIP DEP data from the bottom 260 m of the core have been used to construct rough GI tie points through GI 25a (around 2590 m) as well as evidence of the Eemian-Glacial transition at 2618 m (Stoll et al., 2024). Observations of cm-scale overturning folds, boudin-like structures, and inclined layers with opposing tilts are observed periodically between 1375 and 2121 m, the depth of the deepest linescan image (Westhoff, 2021; Weikusat, 2020). Due to the rough synchronization of DEP data below the depths of the linescan images, these physical observations of cm-scale disturbances are not interpreted as significant breaks in the climate record.

**Dome Fuji, DF2 (Length: 3035.22 m):** The integrity of the Dome Fuji ice core climate record is discussed by the (Dome Fuji Ice Core Project Members, 2017) and summarized here. A chronology for the upper 3028 m of the 3035 m Dome Fuji core was established through the synchronization of $\delta^{18}O$ records to the EDC $\delta D$ profile. Physical observations of inclined layers begin at 2400 m and show distinct stepwise increases in inclination: ~8° between 2450 - 2600, ~20° between 2600 - 2800, ~40° between 2800 - 2900, ~45° at 2950 m, and ~50° at bedrock. Despite the observations of inclined layers, which are attributed to spatially variable basal melt conditions, explicit observations of folded layers were not noted and the synchroneity of the $\delta^{18}O$ and EDC $\delta D$ profiles are considered evidence of an intact climate record within the depths of inclined layers.

**North Greenland Ice Core Project, NorthGRIP2 (Length: 3090 m):** At NorthGRIP, the continuity of the 2544 – 3073 m zone of the 3090 m length core was confirmed by matching NorthGRIP $\delta^{18}O_{atm}$ and $CH_4$ records to EDML and EDML1 chronologies (Capron et al., 2010). Depth shift analysis at 2940 m showed the expected shift between $\delta^{15}N$ and $CH_4$ vs $\delta^{18}O$ during Dansgaard-Oescher (DO) 24, and was used to confirm the continuity of the deepest layers (North Greenland Ice Core

Project Members, 2004). Like at WAIS Divide, small scale stratigraphic disturbances are observed a few hundred meters above
bedrock (Svensson, 2005), but are not considered large enough to impact the continuity of the climate record.
**South Pole Ice Core, SPICEcore, SPC14 (Length: 1500 m | Ice thickness: 2700 m):** Continuity through the end of the core
is established through synchronization of $CH_4$ fluctuations to WAIS Divide ice core (Epifanio et al., 2020). Notably,
SPICEcore drilling stopped 1200 m above bedrock.
**WAIS (West Antarctic Ice Sheet) Divide, WDC06A (Length: 3405 m | Ice thickness: 3455 m):** The continuity of the
WAIS Divide core is confirmed above 2850 m by annual layer counting, and below 2850 m via synchronization of WAIS
Divide $CH_4$ measurements to the NorthGRIP $\delta^{18}O$ record and a refined Hulu Cave speleothem $\delta^{18}O$ record (Buizert et al.,
2015). Notably, the 3405 m WAIS Divide core ends 50 m above bedrock, so continuity in the uncored 50 m basal unit is not
confirmed. Mm-scale or smaller stratigraphic disturbances are observed at 3150 and 3232 m (Fitzpatrick et al., 2014) but are
not considered large enough to impact the continuity of the climate record.

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
