# Peer review of "Advancing interpretation of incoherent scattering in ice penetrating radar data used for ice core site selection"

_EGUsphere, 2024_

## Author Comment (AC1)

Dear Julian,

We appreciate your thorough and thoughtful read of our manuscript. We're glad you found the work informative and are excited to clarify the text and make figure improvements following your suggestions. Here, we *provide a response to each of your comments* and describe anticipated changes to the manuscript.

> Discussion section (Section 4): The authors mentioned early in the paper that the scattering or lack of signal may also be related to data acquisition and processing (see end of Section 2). This is again briefly mentioned in the conclusions. This is something that has repeatedly been on my mind whilst reading this paper, and I wonder whether this could be mentioned/discussed again somewhere in the Discussion section as a caveat. Would the authors expect to see more (or less) scattering if the radar data shown in Figure 3 and 4 be processed differently (i.e. using a homogenous processing workflow from the raw data to all the radar products analysed here – i.e. in a similar way to what the Open Polar Radar project aims to do) and thus, if the authors think that the conclusions drawn from Figure 3 in particular may be potentially different as a result? This is seldom discussed in the paper, but I wonder whether there is scope to add a few sentences on this, and perhaps the discussion section is a good opportunity to add this as a potential caveat.

*For radars of identical design, image processing could significantly affect the way diffuse and specular reflections are expressed in radar imagery (as you rightfully point out). SAR focusing can collapse diffraction hyperbola from diffuse scatterers to point targets, which (if concentrated in ice of a particular age) may look similar to the more specular isochrons. Incoherent and coherent averaging (or "multi-looking") could result in destructive interference of incoherent scattering that reduces the signal to noise ratio. For both qualitative interpretation (like the work done here) and quantitative interpretation (through methods like delay-doppler analysis), data-preprocessing could change the way the data are interpreted, and we will work to incorporate some of those caveats into the discussion.*

*Separate from the way the scattering is interpreted, differences in radar architecture (most notably differences in center frequency and bandwidth) can change the nature of the recorded scattering itself. Apparently coherent, specular horizons can turn into diffuse scatterers as the scale of dielectric heterogeneity (or interface roughness) approaches the frequency-dependent Rayleigh Roughness Criterion. And at some frequency limit, it is likely that all layering within ice sheets will appear as diffuse scattering. What we interpret here is most relevant for the typical ice penetrating radar frequencies and bandwidths. We will also add a caveat to the text to account for expected differences for ultrawideband or ultra high-frequency systems.*

> There is also, of course, the subjectivity in identifying whether the incoherent layering is diffuse, laterally homogeneous, or laterally heterogenous (and how does one set of eye, with one image processed in a certain way to emphasise specific sections or patterns in the ice that may be not be optimised for the type of analysis made in this paper, determines the type of scattering observed as "*strictly*" as it is done in, for example, Figure 4)? I see this paper as a good opportunity to discuss these in some more details, if possible.

*You're absolutely right that the types of qualitative interpretation we do here is subjective – we will try to provide more descriptive language to explain how we did our categorization*

*to try and help future ice core site selectors standardize their practice. But underlying your comment here is a desire for a quantitative framework for layer categorization. We can provide some guidance for how that might work when you have standardized data, which could be a nice contribution for future studies. Thanks for hinting at this, as it provides a clear way to increase the reach of our work.*

**Figures:** Overall, I found the figures very interesting but lacking in clarity or additional information in the text/caption that may help the reader understand them. This is particularly true for Figure 3 (see below for specific comments), which has a lot of information, and the reader is left to do a lot of the work to try to piece together all the information that is being presented. The authors may want to consider whether they could split this figure up into several ones, perhaps ordered by region (Greenland vs Antarctica), make labels and legends bigger and much more simplified, and provide a full caption which may help guide the reader. The other figures also need much better captions to explain the different elements being presented (again see my below comments).

*Structuring the figures was one of the hardest parts of writing this figure for us – we agree that they are information dense, and there may be ways to make them more accessible. See our responses to your specific requests below.*

**Data availability**: There is no mention of where readers can access the data presented in this paper. Could you please add this for all the radar data and other associated datasets presented in this paper?

*We will link to a data repository with all of the radar data used to generate the figures in this study, and in the supplementary material we provide referencing and identifying flight information for each line in its original repository (for those radar images drawn from open access repositories). We will also provide code to reproduce the figures. We were simply waiting to finalize our data repository at the publication stage, but thank you for maintaining accountability for open access, we agree that it is an important attribute of any study.*

**Line Item Comments:**

Line 23-24: "And while […] future ice coring initiatives hope to build…". Confusing grammar, please rephrase

*Changed to: "These cores capture global climate changes over the Holocene and Late Pleistocene (Wolff et al., 2010). Future ice coring initiatives hope to build on that record, both extending it further back in time (Jouzel and Masson-Delmotte, 2010) and measuring regional climate change (Mulvaney et al., 2021) during specific climate periods (Fudge et al., 2023)."*

Line 27: "specific ice" – what is meant here? Replace maybe by "stable" or "climatically stable"?

*The objectives for ice core collection can actually vary quite a bit – scientists could be targeting ice of a particular age, ice that is extremely old, ice that flowed in from a particular region. In this case "specific ice" just means that there are characteristics desired for the ice core acquisition, and site selection needs to be able to identify those characteristics in advance of drilling.*

Line 29: not just "accumulation and ice flow" – add basal melting too

*Content changed to reflect reviewer comment.*

Line 29: Reference to Schroeder et al. 2020 – could add a few more references here. Examples: Bingham et al. 2024 (in review at TC, https://doi.org/10.5194/egusphere-2024-2593); Chung et al. 2023 (https://doi.org/10.5194/tc-17-3461-2023); Karlsson et al. 2018 (https://doi.org/10.5194/tc-12-2413-2018)

*Content changed to include references.*

Line 30: "shallow" – replace by "the upper 2/3" as shallow is an understatement and is also a big vague.

*Content changed to include "the upper three-quarters"*

Line 31: "on what incoherent scattering …" – add "on what incoherent scattering in deep ice" to differentiate with the previous sentence which mentions coherent homogenous layering in the top part of the ice column

*Content changed to include addition of "deep"*

Line 35: I would add a few references to seminal work on this topic in the existing list of reference you provide here. For example, Millar (1982), Hammer (1980), Harrison (1973) works would be great here.

*Content changed to include references.*

Line 39: Add Chung et al. (2023 – DOI already provided here) as an additional reference to Lilien et al. 2021)

*Content changed to include reference.*

Line 39: "16 ice cores" – add "across Antarctica and Greenland"

*Content changed to reflect reviewer comment.*

Line 48: add Bingham et al. 2024 to the Dowdeswell and Evans reference

*Content changed to include reference.*

Line 53: "to an (up to)..." – should be "a". Also please provide a reference to this sentence.

*Because the strength of the fabric controls the bulk permittivity, you can have (a) an isotropic crystal fabric with no dielectric contrasts induced by individual crystal anisotropy, (b) a perfect vertical C-axis maximum that transitions to a perfect horizontal C-axis maximum which would induce a ~1.3% contrast in dielectric permittivity (the same difference that exists between the C-parallel and C-perpendicular axes for individual crystals), or (c) any intermediate contrast between those end members. It is for that reason that we prefer the phrasing (up to) rather than "a" in this sentence, as the magnitude of the fabric induced contrast must fall between*

*~0% and ~1.3%. But we have added a citation to Matsuoka et al., 1997, where the single crystal anisotropy values were measured and published.*

Line 59: add Bingham et al. 2024 to the Fahnestock et al reference already provided.

*Content changed to include reference.*

Line 61: can you provide additional references to the Schroeder et al. 2020 reference here? You provide references to science papers for the previous sentence, but only a review paper for this one. It would help to point the reader to additional science papers that discuss this point.

*Content changed to include references (and some "e.g."s to point out that it is difficult to be comprehensive in referencing these broad study topics).*

Line 64-65: Please refer to Figure 2b here.

*Because we have text that walks the reader through Figure 2 in the following paragraph, we prefer to wait to reference it until we have a more guided introduction to the concept (even though you are right, the text in lines 64-65 is definitely relevant to Figure 2).*

Line 106-108: Add reference here. Perhaps Young et al., 2021 (https://doi.org/10.1029/2020JF006023) is a good starting point.

*Content changed to include reference.*

Lines 189-194: I was confused when reading this paragraph (and some sentences preceding this) about the lack of figures that would illustrate the description of the patterns found at each ice core locations. I think this is because these are not referred to in the text explicitly. I think that mentions of Figures 3 and 4 throughout the text (with sub panels) would help greatly to guide the reader to these figures. As it stands, I read this paragraph but ask myself why there are no figures showing this in the paper, only to find out later that these exist further down the paper but are not being referred to in the text.

*As you point out in your comments on figures, Figure 3 contains a lot of information. We tried to help situate the reader in lines 170-172, suggesting that everything that follows in this section is built on Figure 3. We have modified the text there to be much more explicit that Figure 3 is essential to the following interpretations, with the hopes that readers reference that figure for all of our description.*

Line 204: "time" – what do you mean by this? I think you mean age-depth? Clarify

*Content changed to "where annual layer thickness is."*

Line 210: Again please mention the figures in this sentence and throughout this paragraph

*Because almost every sentence in these paragraphs refers directly to content captured in Figure 3, we prefer the strategy of emphasizing its importance at the start of the section. Hopefully the modified text from lines 170-172 accomplishes your intended goal, but feel free to let us know if you still think it is insufficient.*

Line 239: "doesn't" – replace by "does not"

*Content replaced with "does not"*

Section 3.2: I found this section really interesting – great addition.

*Thank you!*

Line 265-273: I wonder if it would be interesting to show the returned echo power graph of a (or several) trace(s) in a figure (perhaps in a modified version of Figure 3 or 4). This would help counteract the problem with the size and colorscale of the radargrams presented which make it often hard to see the pattern of scattering.

*To counteract the problem with the size, we have increased the size of all of the radargrams in figures 3 and 4.*

Line 326: replace "radar data" with "ice-penetrating radar data"

*Content changed to "ice-penetrating radar data."*

Line 331: replace "to" with "do"

*Content changed to "do."*

Line 334: "time is compressed" – again, the use to the word "time" is maybe a bit confusing to me. Perhaps replace by "age-depth"

*Changed*

Line 335: replace "is" to "to be"

*Content changed to "to be."*

**Figure and captions:**
Figure 1 (caption): Specify what the colormap and reliefs show and where these data come from (also the grounding line and IMBIE drainage catchments please).

*Content changed to include citations.*

Figure 1: you could also add another axis in the "Core length" diagram in Figure 1 which shows the age of each ice cores in combination with the depth axis already provided.

*While we agree that age information might be useful for the reader, the implementation would not be as straightforward as you describe. The depth-age scale for each core is different, which means none of the cores could share an age axis. We've decided to omit this for now, but do provide some age information in the supplemental material.*

Figure 2 (caption): Could you add in the caption where the datasets you present are from (source + radar system type)?

*We will add a reference to Supplementary Table 1, which has the full system characteristics, and fix the typo which currently references Figures 1c instead of Figure 2c.*

Figure 3: I like this figure a lot, however:

The caption does not provide many details that could help guide the reader to each part of the figure (e.g. the left-most plots in each subplot are not explained – are these c-axes plots?). And what about the plots with the green line through them? Perhaps the confusion stems from the fact that there is a lot of information on it, which I don't particularly mind and sometimes I think this is necessary, but it must be properly explained either in the text or in the caption. Having read this multiple times, I am left frustrated that it takes more than a couple of minutes to really get through all the elements presented in the figure.

The "layer slopes" legend is not clear enough and I can't see these very well on the plots

The difference between "no data" and "no visible layering" is too similar and I can't see the difference between the two

The difference between the "+" for the thin and thick sections is not very obvious either. Also, what does "Sampling" mean with regards to these two "+" symbols?

In general, I would say that there is maybe too much information on it, and I would recommend simplifying it a bit but also perhaps making multiple figures from this one, such as by regions or sub-regions. This would also allow for the radargrams to be stretched horizontally a bit so that the patterns are much more visible. Perhaps altering the color scale or adding some gain to the radargrams would also be beneficial, as I'm left having to trust the authors a lot about what they "see", when I can't really see it myself very clearly due to the small size of the figure and the overload of information being presented. This refers also a bit to my general point above with regards to the processing of the radar data that is used to make the interpretations in this paper (of course one could argue this is the case for any dataset, but it would be worth addressing this point in the paper a bit more).

*We agree that this figure became overly complex – a reflection of the fact that we (as authors) wanted to compare all the available data before making conclusions, but that the reader doesn't actually need some of what we present. The Schmidt plots do not add any meaningful information to the figure. We will add idealized Schmidt plots of each of the fabric types to the fabric observations section of the legend instead. Similarly, we will pull out the line scan images from EDML, NEEM, NorthGRIP, & GISP2 into a separate figure that showcases visual observations of the different scales of folded layers. That will leave room for doubling the width of the radargrams without extending the figure across two pages. We will also add grid lines across the figures so that people can more easily see the connections (or lack of connection) between the fabric and layering transitions visualized in the colorbars and changing quality of radar backscatter. To improve color contrast between no data and no visible layering, we can make no data black (which would be consistent with the fabric observations). We have also standardized the depth range across the figures, so they each show ice from 850 m above the bed to 50 m below.*

*We will also update the caption to the following, which provides more context for the reader:*

*"Synthesis of fabric measurements, layering observations, and radargrams at the nine deep ice core drill sites with comprehensive datasets. The left-most section of each ice core panel presents a 1-D scatterplot that marks sampling depths where thin sections (and, where*

*applicable, thick sections) were collected for crystal orientation fabric analysis. \*At GISP2, only some of the sampled thin sections have published data (indicated by the black + symbols), and † at Vostok, the original sampling rate is unpublished, with only a few thin sections and general observations available in the literature. The color bars from each ice core panel summarize fabric observations (left) and layering observations (right) as described in the literature. Fabric observations are simplified into a tripartite classification: strong single maximum (white), weak single maximum or elliptical (cyan), or girdle (green). Other fabric observations or depths with no fabric observations are black.  Where sampling frequency permits, or where gradual fabric transitions are noted in the literature, color gradients are used to represent gradual transitions. The right color bar presents layering observations, with colors reflecting the scale and nature of disturbances: planar, undisturbed layers appear in purple, while progressively disturbed layers are shown in yellow and then red as disturbance size increases to > 50 cm. Sloping layers are indicated by increasing layer slopes (>15°, >30°, >45°). Diffuse layering appears in lavender, and sections with no visible layers appear in grey. Radargrams from each ice core site span ~700-1000 m of the ice column. Fabric observations sourced from: EDML (Eisen et al., 2007; Faria et al., 2018; Weikusat et al., 2013), Dome Fuji (Saruya et al., 2022, 2024), NEEM (Eichler, 2013; Montagnat et al., 2014), NorthGRIP (Wang et al., 2002), Dome C (Durand et al., 2009), GISP2 (Gow et al., 1997), GRIP (Thorsteinsson et al., 1997), Siple Dome (Gow and Meese, 2007), Vostok (Obbard and Baker, 2007). Layering observations sourced from: EDML (Faria et al., 2010, 2018), Dome Fuji (Dome Fuji Ice Core Project Members, 2017), NEEM (Jansen et al., 2015), NorthGRIP (Svensson, 2005), Dome C (Durand et al., 2009), GISP2 (Alley et al., 1995, 1997; Faria et al., 2014; Gow et al., 1997), GRIP (Alley et al., 1995; Dahl-Jensen et al., 1997; Johnsen et al., 1995; Landais et al., 2003), Siple Dome (Gow and Meese, 2007), Vostok (Lipenkov and Raynaud, 2015; Raynaud et al., 2005; Souchez et al., 2002). Radar system characteristics can be found in Supplementary Table 1.“*

Figure 4 (caption): here and in the main text, it would be great if you could refer to your Appendix A, which describes whether a break in the climate record is visible in ice cores and hence can be seen in the radargrams.

*We've included a reference to Appendix A here, and include a reference to it where Figure 4 is introduced (on line 298-299)*

Figure 4: what is the dotted red or black lines in some radargrams (e.g. for GISP2?)

*We have added a row to the legend indicating that the dotted red and black line represents "laterally heterogeneous incoherent scattering visible along extended profile (see figure S3)"*

Figure S2 (caption): there is no "(c)" in the figure, but 2x "(a)"

*Ah, yes, there are actually two letters superimposed within the figure where c is. We've fixed this – thank you for noticing!*

Thank you again for your thoughtful review, we believe the changes made in response to your comments have significantly improved the manuscript.

Ellen + Nick

---

## Author Comment (AC2)

Dear Reviewer,

We appreciate your careful evaluation of our manuscript, and particularly value your suggestions for improvements to the figures. We agree that this manuscript acts as a call to action – there is more work to be done, and we hope this data compilation motivates that work! Here, we *provide a response to each of your comments* and describe anticipated changes to the manuscript.

**Comments**

> Since the figures are essential for the conclusions derived, they should be improved by enlarging them and making the colour bars showing layering and fabric observations as well as scattering information wider. It might be reasonable to use the height of a page in landscape mode as height of a subpanel shown in figures 3 and 4.

*We agree with your assessment that the figures are particularly important in this work, and both you and our other reviewer rightfully point out that we need to de-densify both figures 3 and 4. We intend to modify both figures significantly, with modifications described below.*

*Figure 3: Line scan data from figure 3 has been removed and pulled into a new figure. The Schmidt plots have been removed entirely. The depth axis of each radargram has been adjusted to be the 850 meters above bedrock and 50 meters below bedrock for each radargram. The width of the radargram has been doubled. The symbology of the thin and thick sections has been flipped such that the thin sections are represented by the thin crosses, and the thick sections are represented by thick crosses.*

*Figure 4: The figure has been rotated into landscape and split into two subfigures such that each radargram extends to cover the majority of the height of the page. The depth axis of each radargram has been adjusted to be 1000 m above bedrock and 100 m below bedrock. For cores where the ice thickness is less than 1000 m, the depth axis starts at the surface and extends to 1100 m.*

> L 33: Amend "All radio-wave scattering originates from electrical contrasts." All radio-wave scattering originates from electrical contrasts in conductivity and permittivity

*Content changed to: "All radio-wave scattering in ice originates from dielectric contrasts."*

> L 53: Please provide reference for your statement „… with transitions in c-axis fabric leading to an (up to) 1% contrast in the polarization-dependent bulk permittivity."

*Reviewer 1 also asked about this sentence. We have included a reference, but provide our response to reviewer 1 here for your evaluation as well:*

*Because the strength of the fabric controls the bulk permittivity, you can have (a) an isotropic crystal fabric with no dielectric contrasts induced by individual crystal anisotropy, (b) you can have a perfect vertical C-axis maximum that transitions to a perfect horizontal C-axis maximum which would induce a ~1.3% contrast in dielectric permittivity (the same difference that exists between the C-parallel and C-perpendicular axes for individual crystals), or you can have (c) any intermediate contrast between those end members. It is for that reason that we prefer the phrasing (up to) rather than "a" in this sentence, as the magnitude of the fabric induced contrast must fall between ~0% and ~1.3%. But we have*

*added a citation to Matsuoka et al., 1997, where the single crystal anisotropy value measurements were originally published.*

L 54: reference of inset (i)?:  …(as in the upper half of fig. 2.a, respectively marked example i), …

*Added reference to figure 2.a.i.*

L 68: "Incoherent scattering typically occurs at rough interfaces …"  please define in physical units

*Apparently coherent, specular horizons can turn into diffuse scatterers as the scale of dielectric heterogeneity (or interface roughness) approaches the frequency-dependent Rayleigh Roughness Criterion. And at some frequency limit, it is likely that all layering within ice sheets will appear diffuse. Center frequencies in this study range from 7.5 MHz to 325 MHz, with most data sourced from radar systems with center frequencies between 150 and 200 MHz. Assuming dielectric permittivity of 3.18, most rough interfaces are on the scale of 28 to 21 cm, but extend from 5.6 m to 13 cm. We have added some text to point the reader toward more literature on this subject, because we agree that quantifying what it means to be a diffuse scatterer is important to consider.*

L 77/78: The statement „each pixel typically represents backscattered energy from only a single subsurface target." is only correct if the transmitted signal and the sample interval are short enough to resolve the layering. This is for airborne RES system usually not true.

*You've highlighted an important distinction – by saying "a single target" we did not intend to focus on the scale of the dielectric heterogeneity generating the backscattered energy, but rather the range of angles from which backscattered energy is arriving. That targets have some adjacency as you move along a radar profile for specular layering but not for diffuse scatterers. We've modified that sentence to read: "…each pixel typically represents backscattered energy from only a single direction of arrival."*

L 119 … „within the ~8 cm diameter ice cores." – doesn't make much difference, but isn't the diameter of m

*You are right that the diameter of deep ice cores varies from site to site depending on the drill, so we expanded this to a range of ~8-13 cm, capturing all cores.*

L137: (first occurrence): I suggest to replace NorthGRIP by NGRIP when referring to the ice core.

*We appreciate this comment, and struggled with identifying convention in the literature for the NorthGRIP2 ice core. Because Dahl-Jensen, Svensson et al., 2005, Capron et al, 2010, and Johnsen et al., 2001 each refer to it as "NorthGRIP" our sense is that this is the convention for the literature. But if you can help us identify a clearer convention in the literature, we are open to changing this!*

L 177: (first occurrence): When referring to the ice core drilled at Dome C use EDC, similar to EDML.

*References to Dome C changed to EDC to be consistent with EDML.*

L 331 Correct "… interfaces to not manifest …"  Correct "… interfaces do not manifest …"

*Content corrected.*

L 492 Oatm  atm should be subscript

*Content changed to "$O_{atm}$".*

L 493, 495: ky  kyr

*Content changed to "kyr".*

References, L695-697 Wang et al 2023 not cited in main manuscript

*Reference removed.*

Reference list in the supplement is incomplete:
L55: reference Dansgaard 1982 is incomplete
L60: Eichler 2013: what kind of thesis; how published?

*The Dansgaard reference is now complete. Eichler reference updated to include "Master's Thesis". DOI for Eichler 2013 is not available.*

Fig. 2:
Captions for a-c should be below the figure
Sublabel should be before the respective text (same for other figures)
Labels of examples in 2a are difficult o read and boxes are too small
2c – it should be clearly stated that the two RES sections are not mapped on the same profile

*We have updated the caption such that sublabels precede the text describing them, and to ensure there is a clear statement about the spatial relationship between profiles in 2c. Both profiles and their coordinates will be available in the supplemental information, so a reader can evaluate our own interpretation. We have also increased the box sizes in figure 2a and changed the labeling to make it easier to read.*

Fig. 3:
The sub- panels of the figure should be larger
It seems to be counter-intuitive to use a bold "+" for thin section and a normal sized one for thick sections.
The depth exis should be of equal length for all panels

*There were similar comments made by reviewer 1 – we have made significant changes to the figure to try and simplify and bring forward the key ideas (and make the interpretation by the reader simpler). Among other changes, we have changed the boldness applied to "+" symbols for thin and thick sections, and we have standardized the depth range across the figures, so they each show ice from 850 m above the bed to 50 m below.*

Fig. 4
The sub- panels of the figure should be larger
The depth axis should be of equal length for all panels

*Similar to our modifications of Figure 3, we have expanded Figure 4 to make the radar data easier to interpret by the reader and standardized the depth axis.*

Fig. S2: label (c) is not readable in the figure
 the feature marked with (c), diffuse, banded scattering, looks very much like 2a i coherent scattering  might not be a typical example; please choose a better one

*This is an important observation, and we want to make sure we address it fully. It is the case that some of the bright reflectors within the glacial ice in Greenland (which can be seen in figure 2a) do behave qualitatively differently from the conductivity controlled layering that makes up the Holocene layering. For that reason, we've changed which layer we indicate for 2ai, which was uncritically selected in our original submission. We've improved the color-scale in figure S2 so that it conforms to the standard color ranges used elsewhere, which we hope makes clear why we interpret the layer we see in this image differently. But we also agree there is ambiguity in labeling this feature as an example of diffuse banded scattering and will adjust the text and caption accordingly.*

Fig S3: Even though it is stated that only examples of strong laterally heterogeneous incoherent scattering are indicated in the figure, the choice of marked features, respectively their distribution is odd.

*You're right that the selective sampling of the arrows doesn't properly capture the full distribution of features visible in this figure. We've changed the language of the caption and the way we indicate the regions of heterogeneous incoherent scattering to try and more fully capture what we think is present in the data.*

Thank you again for your thoughtful review, we believe the changes made in response to your comments have significantly improved the manuscript.

Ellen + Nick

---

## Author Response (AR1)

Dear Nanna B. Karlsson,

Thank you for your review of our manuscript and for your additional suggestions for revisions. We have done our best to address each of your comments and the comments from our two reviewers. The result is a significantly improved work, with clearer text and figures, and the addition of cutting-edge results from the EGRIP ice core. You can find a discussion of our changes both here and in our response to reviewers (with our direct response to comments provided in blue).

Thank you once again for your constructive feedback, and we look forward to continuing the evaluation process.

Ellen and Nick

**Comments from the editor:**

I agree with referee #1 that it would be helpful to include some information on age. Consider if the maximum age of the ice core could simply be stated next to the ice core name or at the end of the black column for each ice core?

*We have added a column to Supplementary Table 2 that includes the oldest continuous ice age for each core, the associated depth, and the associated references from the literature.*

Please consider including the EGRIP site. Drilling is now complete, and while we do not have all the associated measurements, numerous studies have already been published based on the ice.

*Thank you for suggesting the addition of EastGRIP to our analysis. Radargrams near the EastGRIP drill site, collected by the 2014 CReSIS MCoRDS system, have been added to Figures 4, 5, and S4. The associated radar metadata has been added to the radar dictionary (https://doi.org/10.7910/DVN/JAQJWZ) and Supplementary Table 1. Published fabric analysis (Stoll et al., 2024) as well as layering observations (Stoll et al., 2023; Westhoff, 2021; Weikusat, 2020) have been integrated into Figure 4. Deformational structures observed in EastGRIP linescan images have been integrated into the text of Section 3: Data and Methods (lines 142-143) as well as Figure 3 (Weikusat, 2020). Assessment of the continuity of the climate record has been added to Appendix A and is based on the GICC05-EGRIP-1 timescale (Mojtabavi et al., 2020) as well as Stoll's preliminary chronology for the remainder of the core (Stoll et al., 2024). A summary of the chemical and physical ice core observations used for these analyses has also been added to Supplementary Table 2.*

*Our discussion of the relationship between fabric observations and incoherent scattering has been added in lines 310 – 319: "At EastGRIP, rapid transitions between vertical girdle and multi-maximum fabrics are observed between 2417 and 2484 m, with a strong multi-maximum fabric established below 2500 m (Stoll et al., 2024). The depth range of the rapid fabric transitions coincides with a layer-conformal package of incoherent scattering. Banding within the package of incoherent scattering is not layer-conformal, and the bands are defined by laterally traceable, abrupt drops in power with depth (rather than laterally traceable, abrupt increases in returned power as we see in the coherent layering above). We describe these traceable lows in power as "nulls", likely the product of destructive interference in scattered energy returning to the radar from multiple directions. The expression of the nulls in the imagery is polarization dependent (Fig. S4; Nymand, 2024 Fig. 3.5) suggesting that this entire scattering package is a result of the fabric." We broadly categorize the quality of radar scattering between 2400-2500 m as laterally heterogeneous incoherent scattering (Figure 5b). However, due to the presence of observed power "nulls," we also include*

*EastGRIP—alongside EDC—as an example where banded but incoherent scattering does not necessarily indicate disturbed basal ice (lines 380–382).*

---

## Referee Report (RR1)

Dear Editor and Authors,

Please find attached my second-round review of Mutter and Holschuh (manuscript number: egusphere-2024-2450) with manuscript title "Advancing interpretation of incoherent scattering in ice penetrating radar data used for ice core site selection".

In general, I would recommend this paper to be published in *The Cryosphere* with *minor* revision, and I very much look forward to seeing the updated version soon. Please find below a series of additional comments.

With best wishes,

Julien Bodart
* * *
**Additional comments**

- Figure 1: Could I suggest an alternative to the "core length" diagram on the right-hand side of Figure 1? I was suggesting in my first review to add the age of the ice cores in a figure in the main text as I think it would help the reader who might not be familiar with the age-depth at each core. The authors mentioned that this information was now in the supplementary materials, which is great. However, I believe that perhaps a good compromise (or opportunity) would be to order the cores in the diagram of Figure 1 by oldest to youngest (or reverse), so 1 in this scenario would be Dome C (currently 9), etc. This is motivated by the fact that there is no current logic behind the ordering of the cores in Figure 1, so ordering them by age would be an easy change that could address my comment (and the editor's).

  Also, and maybe this is unfair at this stage of the manuscript, but do the authors think that the new WACSWAIN (Wolf et al., 2025; https://doi.org/10.1038/s41586-024-08394-w) ice core at the Skytrain ice rise site (WAIS) could be added to the manuscript? There is evidence of disruption in the depth-age scale in the deepest part of the core which remains unresolved. I know this is very early results and comes in late in this paper's process, but perhaps this could be interesting to add if there is time. This is not a strict requirement though and I understand that such addition at this stage is slightly unfair. I leave it up to the authors.
- Line 223: "of to" – rephrase
- Conclusion: I appreciate the observations made in this paper and believe that the conclusions from the analysis of the radar data made here is valid; however, I would like to see another sentence in the conclusion which highlights the potential subjectivity of the analysis and acknowledges that conclusions are not entirely independent from the acquisition or processing of the underlying radar presented here. I leave it up to the Editor and other reviewer(s) whether this step is a pre-requisite for acceptance of the manuscript, but in my opinion, it is an important caveat to add.

  Indeed, if I am an ice-core person with little knowledge of radar, I might assume that this type of analysis/conclusions (especially Figure 5) are independent from things that can affect the quality of the radar data in its current form (i.e. acquisition or processing, whether preprocessing or post-processing to enhance layering for example) and that such strict classification can be made without potential subjectivity, when in fact, and as acknowledge by the authors in their response to the reviews, there is some subjectivity when it comes to this analysis. This is not just subjectivity from the human eye, but also from the pre-processing or acquisition frequencies which affect the strength of the reflectors which are used to make the conclusions in this analysis. I take, for example, Figure 5a "Dye-3" sub-panel, where one could argue that there is some layering below the boundary that the authors put as "no signal", or Figure 5b "Dome F" where different conclusions could be made as to what each section represents. The addition of a simple gain function to these data could likely alter slightly the exact location of these boundaries and the type of scattering they represent, but I believe that the conclusions of the paper do not really acknowledge this caveat clearly.

I think the addition of Lines 125-127 earlier in the manuscript was really useful, thank you for that. But potential ice-core experts might skim through this and focus on the conclusions, and I think it would be worth adding a sentence here to acknowledge this caveat. I do note your addition of paragraph 443-449 in the Conclusion which might be read as an acknowledgement of such caveat, but in my opinion, it lacks a clear acknowledgment of the dependency of acquisition and pre-processing steps which could affect the "strict" classification made in e.g. Figure 5.  A simple sentence, as in Lines 125-127 would be enough for me. As already mentioned, I leave it up to the Editor to decide whether this point is a bit too harsh. In any case, I really enjoyed reading the paper, I learned a lot, and I am sure it will be read by many others in the community for the right reasons, so I thank the authors for their work on this and sharing their findings and insights with the community.

---

## Author Response (AR2)

Dear Editor and Authors,

Please find attached my second-round review of Mutter and Holschuh (manuscript number: egusphere-2024-2450) with manuscript title "Advancing interpretation of incoherent scattering in ice penetrating radar data used for ice core site selection". In general, I would recommend this paper to be published in The Cryosphere with minor revision, and I very much look forward to seeing the updated version soon. Please find below a series of additional comments. With best wishes, Julien Bodart

Julien – Thank you for your careful consideration of our manuscript throughout the review process. Your comments have helped us think deeply about the data and their implications, and have resulted in a much stronger manuscript as a result. For your specific recommendations, see our comments below. Thank you again.

Ellen and Nick

Additional comments:

- Figure 1: Could I suggest an alternative to the "core length" diagram on the right-hand side of Figure 1? I was suggesting in my first review to add the age of the ice cores in a figure in the main text as I think it would help the reader who might not be familiar with the age depth at each core. The authors mentioned that this information was now in the supplementary materials, which is great. However, I believe that perhaps a good compromise (or opportunity) would be to order the cores in the diagram of Figure 1 by oldest to youngest (or reverse), so 1 in this scenario would be Dome C (currently 9), etc. This is motivated by the fact that there is no current logic behind the ordering of the cores in Figure 1, so ordering them by age would be an easy change that could address my comment (and the editor's).

  We understand your desire to add age information to this figure, as it is an interesting dimension one can use to compare cores. We believe the current alphabetical listing makes ice cores more easily findable for the reader, and that additional information in the figure, while interesting, makes it less effective for its primary function – access to the ice thickness and location of each core site (relevant for deep deformation). But, we have constructed the figure as requested and include it in the supplementary figures (Fig. S5) for the manuscript.

  Our earliest draft of the figure included more information about each core. Most notably, it included the timeline for site selection and exploration (as that is what motivated this work originally). In that version (provided below) we found that cross-referencing the figure and the text became more and more complicated, and as the figure got more information dense, the information we cared most about got lost in the noise. In redrafting the figures during the most recent round of review, we considered several options to include the age based on your recommendation, including re-ordering the cores, color-coding their labels based on maximum age, and more. We ultimately concluded that these modifications made it harder for the reader to quickly determine where cores were, the piece of information we care most about here.

[Figure]

- Also, and maybe this is unfair at this stage of the manuscript, but do the authors think that the new WACSWAIN (Wolf et al., 2025; https://doi.org/10.1038/s41586-024-08394-w) ice core at the Skytrain ice rise site (WAIS) could be added to the manuscript? There is evidence of disruption in the depth-age scale in the deepest part of the core which remains unresolved. I know this is very early results and comes in late in this paper's process, but perhaps this could be interesting to add if there is time. This is not a strict requirement though and I understand that such addition at this stage is slightly unfair. I leave it up to the authors.

We spent time reading the literature on the WACSWAIN project, considering what data we would need to meaningfully include it in our analysis. The volume of discontinuous ice in the record there spans a different scale (10s of meters) than we are primarily focused on at the other core sites (100s of meters), and without physical properties measurements published, there is little we can say about the radar data at Skytrain Ice Rise. But your recommendation got us thinking about how we should reference the coastal domes generally: at present, we include Taylor Dome, but not Law Dome, Skytrain Ice Rise, Berkner Island, or the ice cores from James Ross Island and the Fletcher Promontory. These all have thicknesses of less than 1000 m, typically don't have published optical logging data or fabric measurements, and are (on average) much younger than the deep ice cores. In reflecting on what should be included based on your question , we actually concluded that we should probably *remove*  Taylor Dome. It did not currently contribute to our narrative, and by removing it we maintain consistency in our practice excluding Skytrain and other coastal domes in our analysis here.

- Line 223: "of to" – rephrase

Fixed. Thanks for catching this!

- Conclusion: I appreciate the observations made in this paper and believe that the conclusions from the analysis of the radar data made here is valid; however, I would like to see another sentence in the conclusion which highlights the potential subjectivity of the analysis and acknowledges that conclusions are not entirely independent from the acquisition or processing of the underlying radar presented here. I leave it up to the Editor and other reviewer(s) whether this step is a pre-requisite for acceptance of the manuscript, but in my opinion, it is an important caveat to add.

  Indeed, if I am an ice-core person with little knowledge of radar, I might assume that this type of analysis/conclusions (especially Figure 5) are independent from things that can affect the quality of the radar data in its current form (i.e. acquisition or processing, whether pre or post-processing to enhance layering for example) and that such strict classification can be made without potential subjectivity, when in fact, and as acknowledge by the authors in their response to the reviews, there is some subjectivity when it comes to this analysis. This is not just subjectivity from the human eye, but also from the pre processing or acquisition frequencies which affect the strength of the reflectors which are used to make the conclusions in this analysis. I take, for example, Figure 5a "Dye-3" sub panel, where one could argue that there is some layering below the boundary that the authors put as "no signal", or Figure 5b "Dome F" where different conclusions could be made as to what each section represents. The addition of a simple gain function to these data could likely alter slightly the exact location of these boundaries and the type of scattering they represent, but I believe that the conclusions of the paper do not really acknowledge this caveat clearly.

  I think the addition of Lines 125-127 earlier in the manuscript was really useful, thank you for that. But potential ice-core experts might skim through this and focus on the conclusions, and I think it would be worth adding a sentence here to acknowledge this caveat. I do note your addition of paragraph 443-449 in the Conclusion which might be read as an acknowledgement of such caveat, but in my opinion, it lacks a clear acknowledgment of the dependency of acquisition and pre-processing steps which could affect the "strict" classification made in e.g. Figure 5. A simple sentence, as in Lines 125-127 would be enough for me. As already mentioned, I leave it up to the Editor to decide whether this point is a bit too harsh. In any case, I really enjoyed reading the paper, I learned a lot, and I am sure it will be read by many others in the community for the right reasons, so I thank the authors for their work on this and sharing their findings and insights with the community.

  We agree that this caveat is important for all readers to consider, and should exist in a prominent place in text. We've updated the paragraph starting on line 443 to make this point explicit, and added a sentence pointing folks to a more objective approach that (while not possible with all of the historical data) *should* be used for future studies of this type.

  Thank you again for all of your work in reviewing this, we really appreciate it.

---

## Author Response (AR3)

Public justification (visible to the public if the article is accepted and published):
Dear Ellen Mutter and Nicholas Holschuh,
Thank you for addressing the comments of the referee so carefully. I am happy to recommend your manuscript for publication pending a few technical corrections that I outline below. Your study highlights the importance of radar data analysis and the comprehensive overview and analysis you have performed will undoubtedly be of use to deep ice core projects in the future.
Best,
Nanna B. Karlsson

*Dear Nanna,*
*We are thrilled to have the manuscript recommended for publication and thank you for your detailed and thoughtful review of our work. Below we have responded to the few technical corrections. Thank you again.*

*Ellen and Nick*

Additional private note (visible to authors and reviewers only):
Caption figure 1: Given the order of the maps, it would make more sense if the caption reads: "Surface elevation maps of Greenland (Porter et al., 2018) and Antarctica (Howat et al., 2019)...", i.e. that Greenland is mentioned first.

*Caption text has been reordered accordingly.*

Line 129: "...collected with radar hardware typical of the earth 2000's" Is this a typo? do you mean "early 2000's"?

*Corrected to "early." Thank you for catching this typo.*

Figure 3 caption: "Ice without layer structure can be due clear ice..." Missing a "to"

*Fixed. Thank you!*

Line 277: "the c-axis fabric transition" -> "the c-axis fabric transitions"

*Fixed. Thank you!*

Fig. 3, 4 and 5: I know this is a bit late in the process but I worry that the purple colours used in Figs. 3 and 4, and the green colour used in Fig. 5 is not colourblind-friendly. For the latter, the contrast with the red colour might not be adequate; for the other two figures, I am worried about the contrast between the purples. Could I ask you to please check this (if you haven't already)? For example, using https://www.color-blindness.com/coblis-color-blindness-simulator/

*Thank you for raising concerns about the visual accessibility of our color schemes. We have reviewed Figures 3, 4 and 5 using Adobe Illustrator's protanopia-type and deuteranopia-type color blindness filters. Based on this review, we believe the purple shades used in Figures 3 and 4 maintain sufficient contrast under both protanopia (red-blind) and deuteranopia (green-blind) filters (see figures below). To enhance contrast in Figure 5, we replaced the red color with a lighter blue. Additionally, in Figure 4, we darkened both the cyan and red shades for better visibility.*

*Below, we provide side-by-side visualization of the original figures in full RGB, protanopia, and deuteranopia color blindness filters, as well as the updated versions of Figure 4 and 5b. While some overlap remains in Figure 4 between the color hues used for fabric observations and those for layering observations under colorblindness filters, we believe the color scheme remains accessible because these hues are displayed in separate columns.*

*Figure 3:*

*Original*         *Protanopia Filter*        *Deuteranopia Filter*

[Figure]

**Figure 4:**

*Original*                    *Updated*

[Figure]

*Figure 4 cont.:*

*Original with Protanopia Filter*     *Updated with Protanopia Filter*

*Figure 4 cont.:*

*Original with Deuteranopia Filter*

*Updated with Deuteranopia Filter*

*Figure 5b (no adjustments needed for 5a):*

[Figure]

*Figure 5b cont.:*

[Figure]